# Entanglement in photoionisation reveals the effect of ionic coupling in attosecond time delays

Ioannis Makos [1], David Busto [1,2], Jakub Benda [3], Dominik Ertel[1], Barbara Merzuk[1], Benjamin Steiner[1], Fabio Frassetto [4], Luca Poletto[4], Claus Dieter Schröter[5], Thomas Pfeifer [5], Robert Moshammer[5], Serguei Patchkovskii[6], Zdeněk Mašín[3] & Giuseppe Sansone [1,7] ✉

Attosecond photoelectron interferometry, based on the measurement of photoelectron spectra generated by a two-colour field, provides access to the photoionisation dynamics of quantum systems. In general, due to the entanglement between the wave function of the emitted photoelectron and that of the parent ion, the dynamics driven by the infra-red field in the photoion can affect the properties of the photoemitted electronic wave packet, when the measurement protocol corresponds to the projection of the total time-dependent wave function onto a specific final state of the bipartite system. This is particularly relevant for molecules, due to their rich internal electronic and vibrational energy structure. Here we show how the polarisation of the ion influences the photoionisation dynamics by introducing an additional time delay in the photoelectrons emitted from $CO_2$ molecules. The delay stems from the entanglement between the photoion and the photoelectron created in the photoionisation process.

The emission of an electron after the absorption of a photon is a fundamental manifestation of the quantised nature of light-matter interaction[1]. With the advent of attosecond technologies[2,3], time-resolved information about the electronic dynamics in the infra-red (IR) assisted photoionisation process has been achieved by extreme ultraviolet (XUV) trains[4,5] or isolated attosecond pulses[6,7]. In particular, the effect of the anisotropic molecular potential[8,9], the influence of molecular shape resonances[10,11], and the nuclear dynamics in isotopologues[12] have been accessed by the reconstruction of attosecond beating by interference of two-photon transitions (RABBIT) technique[4]. This technique is extremely sensitive to the electronic structure of the target systems and, in particular, to the presence of resonances such as autoionising resonances[13,14], shape resonances[15,16],

and shake-up states[17]. The interferometric nature of the RABBIT technique provides a high temporal resolution that has been used to resolve in time the formation of Fano profiles[18] and to characterise the role of Cooper minima in attosecond time delays[19]. As such, this approach can be used to benchmark theoretical models describing the ultrafast response of correlated multielectron systems to an external light field.

In atoms, the attosecond time delays in photoionisation can usually be decomposed into two terms associated with the photo-ionisation process determined by the XUV field alone (a term referred to as the Wigner time delay)[20,21] and one due to the presence of IR radiation (referred to as the continuum-continuum delay)[22,23]. Recently, it has been shown that the decomposition holds for

¹Institute of Physics, University of Freiburg, Hermann-Herder-Straße 3, Freiburg, Germany. ²Department of Physics, Lund University, PO Box 118, Lund, Sweden. ³Institute of Theoretical Physics, Faculty of Mathematics and Physics, Charles University, V Holešovičkách 2, Prague 8, Czech Republic. ⁴Istituto di Fotonica e Nanotecnologie, CNR, via Trasea 7, Padova, Italy. ⁵Max-Planck-Institut für Kernphysik, Heidelberg, Germany. ⁶Max-Born Institute, Max-Born-Str. 2A, Berlin, Germany. ⁷Freiburg Institute for Advanced Studies (FRIAS), University of Freiburg, Albertstraße 19, Freiburg, Germany. ✉e-mail: giuseppe.sansone@physik.uni-freiburg.de

photoelectrons characterised not only by high but also by low kinetic energy[24]. In the case of photoionisation of molecules, the separation of the two terms is not straightforward, and additional terms have been shown to play an important role. The polarisation of the molecular ion, due to the presence of an intrinsic dipole moment, has been predicted to introduce an additional time delay in photoionisation[25–27]. Furthermore, polarisation effects induced by the IR field in neutral molecules affect the photoionisation yield triggered by an attosecond pulse train[28].

In general, the photoionisation process resulting from the absorption of an XUV photon leaves the target in an entangled state in which the photoelectron wave functions of different energies are associated with the corresponding ionic states[29]. Recent theoretical[30] and experimental[31] works have shown that the combination of two time-delayed attosecond pulse trains can control the degree of entanglement and coherence in the photoionisation of molecules. In atoms, the entanglement between photoelectrons and ions can be controlled by varying the bandwidth of XUV pulses[32]. The quantum state of a photoelectron generated by attosecond pulse trains has been characterised using different approaches[33,34]. The interaction of an intense laser field with an atomic target has been investigated, focusing on the entangled nature of the photoelectron released in the process of nonsequential double ionisation[35]. Strong-field interactions have also been proposed as an experimental platform for the implementation of a Bell test of quantum entanglement in atomic photoionisation using circularly polarised radiation[36]. Moreover, intense laser fields can be used to prepare and probe on ultrafast timescales the entanglement between atoms resulting from molecular dissociation[37].

The role of entanglement in photoionisation has also recently been investigated in the interaction of atoms with intense XUV fields[38,39], showing, for example, that XUV two-colour fields can be used to control the degree of coherence of the photoelectron wave packet emitted by the interference of linear and nonlinear photoionisation processes[40].

In molecules, the control of the emission of entangled electrons has been recently reported[41]. Recent theoretical analysis has predicted that IR absorption in the molecular cation can lead to transitions between the entangled components of the cation-photoelectron wave function, resulting in an additional attosecond delay term[42]. Laser-driven dipole transitions between different molecular ionic states were first investigated in the context of multichannel high-order harmonic generation in the $CO_2$ molecule[43], elucidating the strong-field driven hole dynamics using an ab-initio approach[44].

Here, we present experimental data obtained in $CO_2$ molecules, which provide evidence for the role of the ion transitions on the photoionisation time delay of the photoelectron wave packet emitted in the two-colour field. The experimental data support the conclusion that the ionic transitions introduce a negative delay in the photoionisation process due to the dipole coupling between two states of the cation induced by the IR field. The entangled nature of the photoion-photoelectron system provides insight into the dynamics taking place in the bipartite system by observing the properties of the photoelectron.

## Results

The experimental setup is shown schematically in Fig. 1a. Trains of attosecond pulses were generated in the spectral range between 20 and 50 eV by high-order harmonic generation (HHG) in krypton (see Supplementary Information (SI) and see Supplementary Fig. 1). The XUV radiation was focused in the interaction region of a photoelectron-photoion coincidence spectrometer (reaction microscope, ReMi)[45–47] by a toroidal mirror. A synchronised IR field was overlapped using a system of two drilled plates, which ensures excellent interferometric stability over several days[48,49]. The RABBIT traces were measured in a mixture of randomly oriented $CO_2$ molecules and

argon; the latter was used to estimate the attosecond chirp of the XUV pulse train[50] (see SI 1.1).

In the spectral range spanned by the attosecond pulse trains, single XUV photoionisation of neutral $CO_2$ results in the generation of the molecular ion in its ground ($X^2\Pi_g$) and lower excited ($A^2\Pi_u$, $B^2\Sigma_u^+$ and $C^2\Sigma_g^+$) states[51], as schematically shown in Fig. 1b. It is important to note that the energy difference between the binding energies of the $B^2\Sigma_u^+$ and $C^2\Sigma_g^+$ states ($\Delta$) is about 1.3 eV, close to the photon energy of the IR pulse used in the experiment ($\approx 1.21$ eV). The Dyson orbitals for the two excited ionic states are shown in Fig. 1b. Additional fragments are measured and resolved in the experiment according to their time-of-flight (see Supplementary Fig. 2).

Different photoionisation channels contribute to the total photoelectron spectrum determined by the absorption of a single XUV photon, as shown in Fig. 1c. The contribution of the different channels is highlighted by dotted and dashed vertical lines. In particular, the position of the photoelectron peaks associated with a molecular ion in the $B^2\Sigma_u^+$ state is marked by green dash-dotted lines. The blue shaded areas indicate regions with an energy width of 300 meV, centred around kinetic energies corresponding to the additional exchange of an IR photon. In a two-colour photoionisation process, these energies would correspond to the central energy of the sideband photoelectron associated with the $B^2\Sigma_u^+$ state. These energy regions are characterised by deep minima in the XUV-only photoelectron spectra, indicating that the sideband signal associated with a cation left in the $B^2\Sigma_u^+$ state can be isolated from the other contributions by restricting the analysis only to the regions corresponding to the minima. This conclusion is further confirmed by the analysis of the amplitude and phase of the $2\omega_{IR}$ oscillation ($\omega_{IR}$ indicates the angular frequency of the IR field) as a function of the photon energy (see Supplementary Fig. 3).

Figure 2a–c shows the two-colour photoelectron spectrum as a function of the relative delay $\Delta t$ between the XUV and IR fields for three different integration angles around the common polarisation direction of the two fields. The sideband photoelectron spectra exhibit periodic oscillations, which are described by the equation:

$$I^{SB}(\Delta t; e) = A_0 + A_2 \cos(2\omega_{IR}\Delta t - \Delta\varphi_{atto} - \Delta\varphi_{mol.}), \quad (1)$$

where $A_0$ is an offset, $A_2$ is the amplitude of the oscillations, and $e$ is the kinetic energy of the photoelectron. The phase of the sideband oscillation can be decomposed into a term related to the phase difference of the two adjacent harmonics of the XUV comb ($\Delta\varphi_{atto}$) and a term introduced by the two-colour photoionisation process ($\Delta\varphi_{mol.}$), which depends on the specific target.

The analysis of the amplitude and phase of the $2\omega_{IR}$ oscillation allows one to isolate the contribution of the sidebands associated with the remaining ion in the $B^2\Sigma_u^+$ state (see SI 1.2). The attosecond time delays $\tau_B$ extracted from the photoelectrons measured in coincidence with $CO_2^+$ corresponding to the $B^2\Sigma_u^+$ state are shown in Fig. 2d. The delay was determined from the phases of the corresponding sideband oscillations according to the equation:

$$\tau_B = \frac{\Delta\varphi_{atto} + \Delta\varphi_{mol.}}{2\omega_{IR}} \quad (2)$$

The experimental data clearly show a variation with the photoelectron kinetic energy due to the attosecond chirp[50] and the photoionisation molecular phase. Moreover, the measurements show a dependence of the RABBIT delays on the angular integration interval. For photon energies higher than $\approx 24$ eV, the curves are shifted by an almost constant offset with respect to the one corresponding to the integration interval ($0° – 90°$).

The phase difference between successive harmonics due to the attochirp ($\Delta\varphi_{atto}$) was characterised and corrected for using photoelectron spectra measured in coincidence with singly charged argon

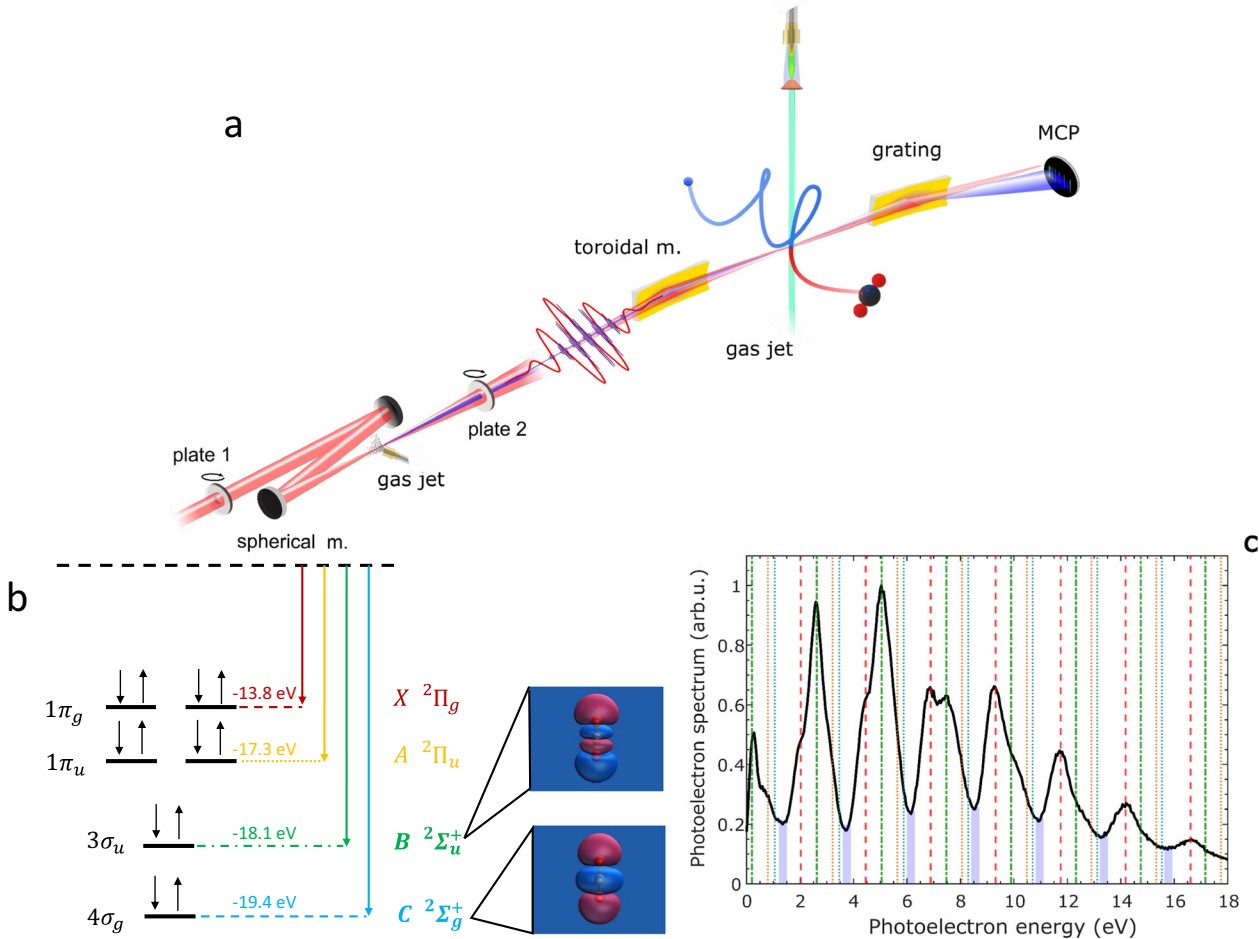

**Fig. 1 | Experimental setup and XUV spectroscopy of CO₂ molecules.**
**a** Schematic view of the experimental setup used for the measurement of RABBIT spectrograms in a mixture of randomly aligned $CO_2$ molecules and argon atoms (MCP: microchannel plate). **b** Electronic configuration of $CO_2$ and binding energies of the first four cationic states together with the Dyson orbitals for the $B^2\Sigma_u^+$ and $C^2\Sigma_g^+$ cationic states. **c** XUV-only spectra measured in coincidence with the parent cation $CO_2^+$ for the integration angle (0° – 90°) along the polarisation direction of

the XUV field. A 200-nm-thick aluminium filter was used to filter out the IR radiation. The vertical lines indicate the expected positions of the photoelectron peaks associated with a cation in the $X^2\Pi_g$ (dashed red lines), $A^2\Pi_u$ (yellow dotted lines), $B^2\Sigma_u^+$ (green dash-dotted lines), and $C^2\Sigma_g^+$ (blue dotted lines) state. The shaded blue areas indicate the expected energy ranges for the sidebands associated with an ion in the $B^2\Sigma_u^+$ state. The binding energies of the first ionic states are taken from refs. 62,63.

ions under the same experimental conditions (see Supplementary Fig. 4). From these data, the attosecond time delay $\tau_{Ar}$ was measured for different integration angles (see Supplementary Fig. 5).

We calculated the expected RABBIT sideband delays at centres of the sidebands, $\tau_B = \arg[T_+^* T_-]/2\omega_{IR}$, from ab-initio two-photon ionisation amplitudes $T_\pm$ for the XUV ± IR pathways using the stationary molecular above-threshold multi-photon R-matrix method[52] for the equilibrium geometry of the neutral molecule. The simulation assumed a monochromatic IR field with wavelength 1030 nm (energy 1.2 eV). Several different molecular models are discussed in this article to analyse the role of the IR-driven dipole transition between the $B^2\Sigma_u^+$ and $C^2\Sigma_g^+$ residual ion states: (a) "static exchange (SE) model", which represents the ionisation in the Hartree-Fock single-channel picture without the possibility of transition between distinct residual ion states; (b) "closed coupling (CC) model", which uses a high-quality quantum-chemical description of the molecule and couples hundreds of residual ion channels[42,53], allowing for all relevant dipole transitions; (c) "CC model without B–C", which is identical to the previous one except that the transition dipole connecting the $B^2\Sigma_u^+$ and $C^2\Sigma_g^+$ states of the residual cation is manually set to zero. Both CC models simulate electronic correlation to a high accuracy in the initial bound state as

well as in the final continuum states. The adequacy of the employed R-matrix model has been demonstrated in the theoretical treatment of one-photon ionisation[53,54], where it yielded photoionization cross sections in good agreement with measurements, including the shapes of major resonances. The difference between the two models is in the transition from the XUV-induced intermediate continuum state to the final continuum state, which can be, in principle, mediated by IR-photoelectron interaction as well as by IR-ion interaction. The second model removes the possibility of the IR-induced transition between the $B^2\Sigma_u^+$ and $C^2\Sigma_g^+$ states. The reference RABBIT sideband delays in argon ($\tau_{Ar}$) were obtained in the same way, using a large polarisation-consistent coupled Hartree-Fock model[55]. Using a procedure described in a previous work[42], the dense structure of the autoionising resonances was smoothed out from the CC calculations.

## Discussion

Figure 3 shows the comparison between the delay difference $\tau = \tau_B - \tau_{Ar}$ for the simulated data and for the experimental measurements. The theoretical models predict significant differences for the attosecond time delays in photoionisation, depending on the included effects. In particular, the inclusion of ionic coupling in the full model determines a

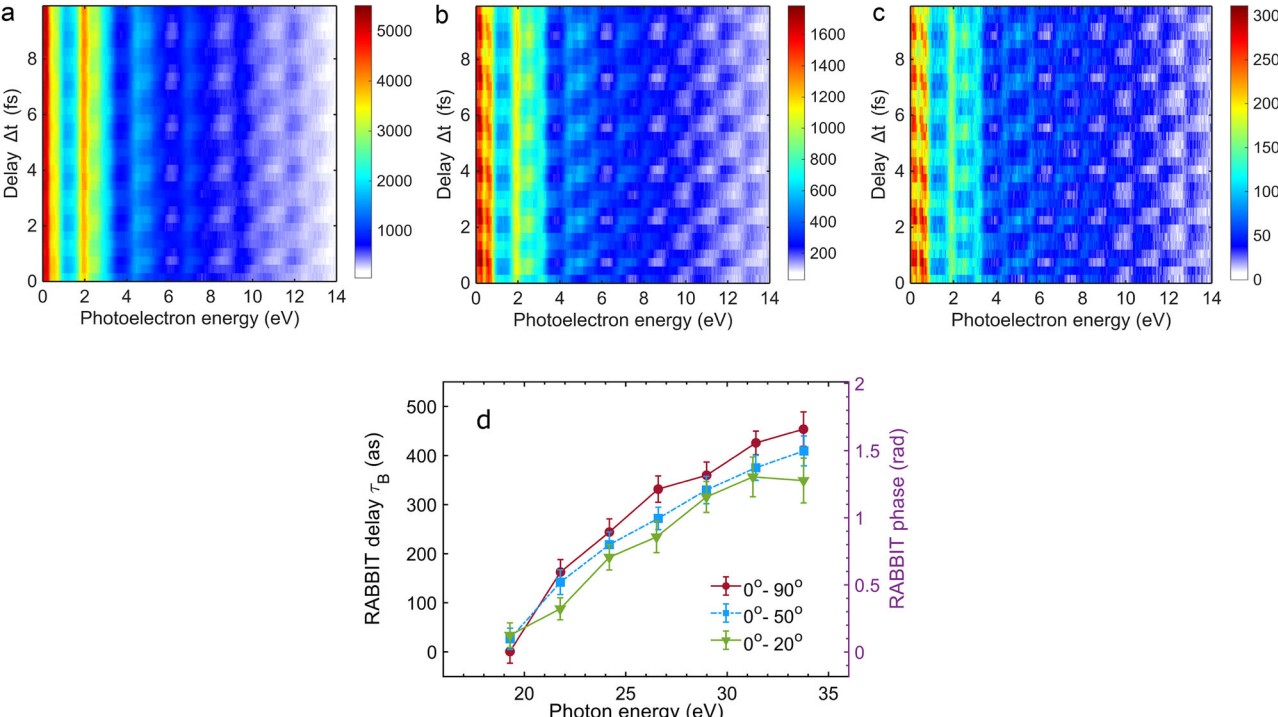

**Fig. 2 | Angle-resolved RABBIT spectrograms in the laboratory frame.** RABBIT spectrograms measured in coincidence with the parent ion $CO_2^+$, considering photoelectrons emitted for angles $\theta$ in the interval $(0° - 90°)$ (**a**), $(0° - 50°)$ (**b**) and $(0° - 20°)$ (**c**) in the laboratory frame. The common polarisation direction of the XUV and NIR pulses corresponds to $\theta = 0°$. **d** RABBIT delay $\tau_B$ extracted from the

RABBIT spectrograms for the different angle integration ranges shown in panels (**a**), (**b**), and (**c**). The first experimental point for the interval $(0° - 90°)$ was set to zero. The error bars (one standard deviation) were derived from fitting the periodic oscillations of the photoelectron spectra using Eq. (1).

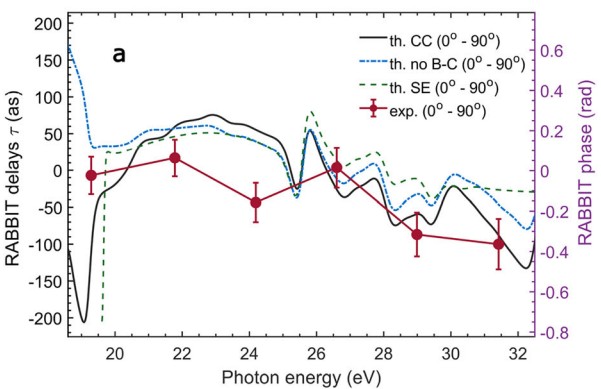

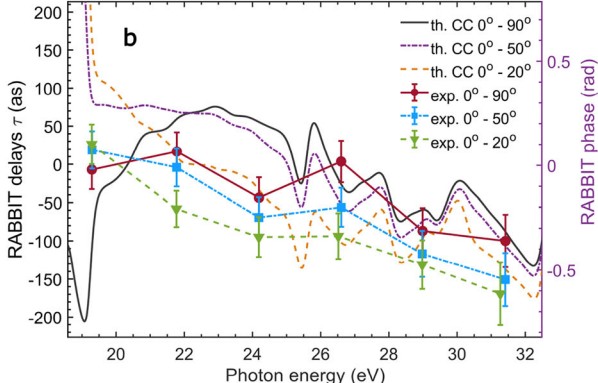

**Fig. 3 | Comparison experiment-theory of the photoionisation time delays.**
**a** Experimental (red circles and lines) difference of the photoionisation time delays $\tau = \tau_B - \tau_{Ar}$ measured in coincidence with $CO_2^+$ and $Ar^+$ integrated over the photoelectron emission angle $(0° - 90°)$. The error bars (one standard deviation) were derived from fitting the periodic oscillations of the photoelectron spectra using Eq. (1). Same difference of the photoionisation time delays $\tau$ extracted from the theoretical models using the static exchange approach (green dashed line), the full model without coupling in the ionic states (blue dashed-dotted line), and the full

model including the ionic coupling (black solid line). **b** Experimental and simulated difference of photoionisation time delays $\tau$ for different integration angles in the laboratory frame: $(0° - 90°)$ (black line for theory, red circles and line for experiment), $(0° - 50°)$ (purple dashed-dotted line for theory, blue squares and line for experiment), and $(0° - 20°)$ (orange dashed line for theory, green triangles and line for experiment). Here, all theoretical lines correspond to the full model. The structures visible in the simulated RABBIT delays $\tau$ in the energy range $\approx 25–27$ eV are due to the $Ar^*(3s^13p^64p^1)$ and $Ar^*(3s^13p^64s^1)$ resonances in argon.

downshift of the photoionisation time delays with respect to the prediction of the CC model without B–C coupling of about 30 as in the photon energy range 26–32 eV (Fig. 3a). In this range, the error bars of the experimental data allow one to distinguish between the different theoretical models, indicating that the experimental results are in good agreement with the predictions of the full model.

The agreement with the full model is further supported by the analysis of the evolution of the photoionisation time delays for different angle integration intervals around the laser polarisation in the

laboratory frame, as shown in Fig. 3b. By reducing the integration angle, the experimental photoionisation time delays shift towards negative values; the same trend can be observed for the full model simulations in the range 26–32 eV. The comparison between the experimental points and the two theoretical models with and without B–C coupling for the integration angles $0° – 20°$ and $0° – 50°$ is presented in the Supplementary Fig. 6.

Only the experimental point around 24 eV in Fig. 3a deviates from the simulations performed with the full model. We attribute the offset

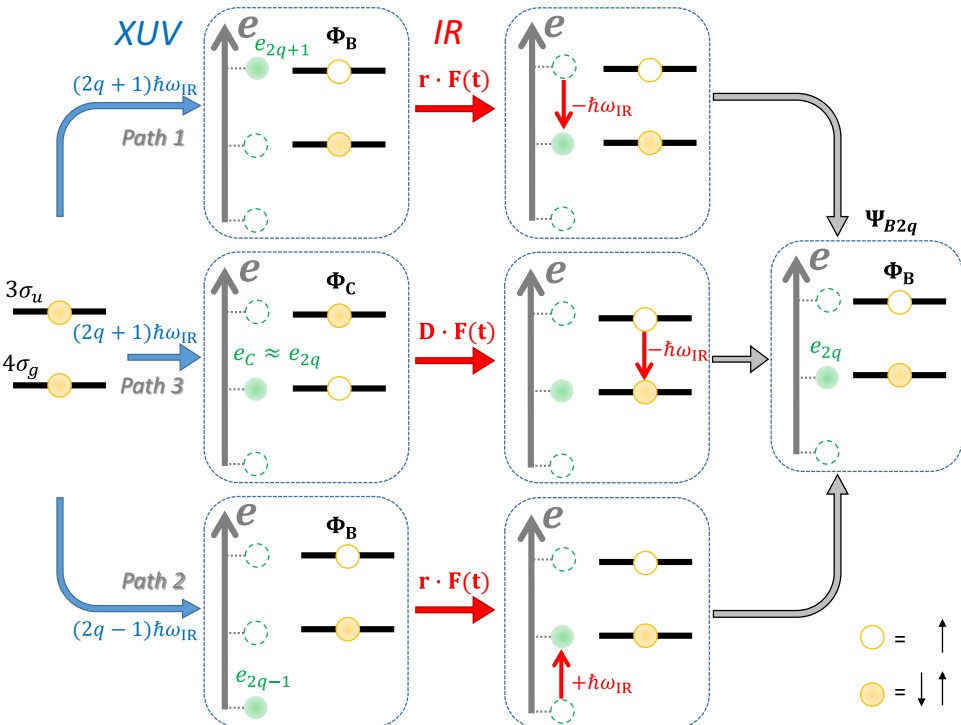

**Fig. 4 | Three-path RABBIT measurement and effect of the coupling between the ionic states on the entangled photoelectron-photoion wave function.** Schematic representation of the two-colour photoionisation process on the $CO_2$ molecule described by the two states $B^2\Sigma_u^+$ and $C^2\Sigma_g^+$. The absorption of an XUV photon of the harmonic $2q \pm 1$ determines a state described by an ion in the $B^2\Sigma_u^+$ state ($\Phi_B$) and a photoelectron with energy $e_{2q\pm1} = (2q \pm 1)\hbar\omega_{IR} - I_p(B)$ (path 1 for $2q + 1$ and path 2 for $2q - 1$, respectively). The subsequent interaction with the IR

leads to the emission (absorption) of an IR photon for the path 1 (path 2), resulting in the final state $\Psi_{B2q}$. The absorption of a photon of the harmonic $2q + 1$ can also lead to the emission of a photoelectron $e_C = (2q + 1)\hbar\omega_{IR} - I_p(C) \approx e_{2q}$ and an ion initially in the $C^2\Sigma_g^+$-state ($\Phi_C$) (path 3). The interaction with the IR field determines an ionic coupling that leads to the transition from the $C^2\Sigma_g^+$ to the $B^2\Sigma_u^+$-state, thus reaching the same final state as the other two paths.

of this point to a small shift of the positions of autoionizing resonances in $CO_2$ and Argon with respect to their accurate values. See SI 1.3 and Supplementary Fig. 7 for a detailed discussion.

The origin of the shift in the attosecond time delays due to the inclusion of the ionic coupling between the $B^2\Sigma_u^+$ and $C^2\Sigma_g^+$ ionic states can be understood by considering the entangled nature of the photoelectron-photoion system and the action of the IR field on both the photoelectron and the photoion (see SI 1.3 and Supplementary Fig. 8).

The oscillations of the sideband yield in a RABBIT measurement are usually described as the result of the interference between two pathways connecting the ground state of the target system to the same ionic and photoelectron final state. For the sideband of order $2q$ and for the final ionic states $B^2\Sigma_u^+$, these two paths (paths 1 and 2) are shown in Fig. 4. In the first path (path 1), a photon of the harmonic $2q + 1$ is absorbed leading to the ionic state $B^2\Sigma_u^+$ (wave function $\Phi_B$; see also Fig. 1b) and a photoelectron with energy $e_{2q+1} = (2q + 1)\hbar\omega_{IR} - I_p(B)$, where $I_p(B)$ indicates the ionisation potential of the $B^2\Sigma_u^+$ state. In the interaction with the IR field, the photoelectron emits an IR photon ($- \hbar\omega_{IR}$), determining a transition between two states in the continuum through the coupling term $\mathbf{r} \cdot \mathbf{F}(t)$, where $\mathbf{r}$ is the one-electron dipole moment operator in the length gauge and $\mathbf{F}(t)$ is the time-dependent IR field. As a result, the final state of the system (wave function $\Psi_{B2q}$) is characterised by an ion in the $B^2\Sigma_u^+$ state and a photoelectron with energy $e_{2q} = 2q\hbar\omega_{IR} - I_p(B)$.

The same final state can be reached by the second path (path 2), characterised by the absorption of a single XUV photon of the harmonic $(2q - 1)$, which releases a photoelectron with energy $e_{2q-1} = (2q - 1)\hbar\omega_{IR} - I_p(B)$, and the absorption of an additional IR photon ($+ \hbar\omega_{IR}$) by the photoelectron, still due to the coupling term $\mathbf{r} \cdot \mathbf{F}(t)$.

The interference of these two paths introduces a delay in photoionisation, which we denote as $\tau_e$.

Due to the interaction of the IR field with the residual ion, a third path can contribute to the same final state (path 3 in Fig. 4). By absorption of a single XUV photon with energy $(2q + 1)\hbar\omega_{IR}$, the cation can be left in the $C^2\Sigma_g^+$ state (wave function $\Phi_C$), leading to a photoelectron with energy $e_C = (2q + 1)\hbar\omega_{IR} - I_p(C)$, where $I_p(C)$ is the ionisation potential of the $C^2\Sigma_g^+$ state. Because the difference between the ionisation potentials of the two states is close to the energy of the IR photon ($I_p(C) - I_p(B) \approx \hbar\omega_{IR}$ (see Fig. 1b)), the photoelectron energy for this path corresponds to the one of the final state $\Psi_{B2q}$ ($e_C \approx e_{2q}$). For this path, the interaction of the IR pulse takes place between the two ionic states mediated by the term $\mathbf{D} \cdot \mathbf{F}(t)$ (where $\mathbf{D}$ is the dipole moment operator in the residual ion), inducing the emission of an IR photon and a transition from the $C^2\Sigma_g^+$ to the $B^2\Sigma_u^+$ state ($\Phi_C \rightarrow \Phi_B$), without modifying the energy of the photoelectron. As a result, the inclusion of the effect of the IR on the ionic part of the wave function introduces an additional pathway leading to the same final state.

According to the previous discussion and from Fig. 4, the system is ionised either by the harmonics $2q - 1$ or $2q + 1$, corresponding to the states of the bipartite system:

$$\Psi_{2q+1} = \underbrace{\sqrt{b_+}\,\psi_{2q+1}\Phi_B}_{\text{path 1}} + \underbrace{\sqrt{c_+}\,\psi_{2q}\Phi_C}_{\text{path 3}} \tag{3}$$

$$\Psi_{2q-1} = \underbrace{\sqrt{b_-}\,\psi_{2q-1}\Phi_B}_{\text{path 2}} + \sqrt{c_-}\,\psi_{2q-2}\Phi_C, \tag{4}$$

where $\psi_{2q-2}$, $\psi_{2q\pm1}$, $\psi_{2q}$ are electronic wave functions with energies $e_{2q-2}$, $e_{2q\pm1}$ and $e_{2q} \approx e_C$, respectively. The quantities $b_\pm$ and $c_\pm$ describe the initial population of the four states of the total system and depend on the photoionisation cross sections and on the intensity distribution of the XUV spectrum. The photoionisation phases and the phases of the XUV harmonics are not expressed explicitly, but are included directly in the wave functions for simplicity. The correspondence between the different terms and the initial state of the three paths indicated in Fig. 4 is indicated below each term. The last one, upon exchange of a single IR photon, cannot lead to the final state $\Psi_{B2q}$ and is not relevant for our discussion. The total wave function of the system is given by the sum of Eqs. (3) and (4). The coupling delay depends on the purity of the reduced density matrix of the state $\Psi_{2q+1}$, which is given by $\gamma = (b_+^2 + c_+^2)/(b_+ + c_+)^2$ (see SI 1.3), because its components contribute to the different interference pathways building the sideband signal.

We note that photoionisation from the $B^2\Sigma_u^+$ state does not lead to molecular dissociation[56], while the $C^2\Sigma_g^+$ state completely predissociates[57]. The combination of these two observations ensures that the coincidence measurements of the photoion $CO_2^+$ with the photoelectrons in the energy range around the expected values of the sidebands associated with the $B^2\Sigma_u^+$ state (see Fig. 1c) correspond to a projection of the total wave function of the system onto the final state of the bipartite system $\Psi_{B2q}$.

The interference of path 3 with paths 1 and 2 introduces an additional time delay $\tau_{ion}$, determining a RABBIT delay given by the sum of two terms:

$$\tau_B = \tau_e + \tau_{ion} \qquad (5)$$

Equation (5) is consistent with the shift between the attosecond time delays predicted by the CC and the CC without B−C coupling models, in which the ionic coupling mechanism is switched off, observed in Fig. 3a (see also Supplementary Fig. 6). While ion-photoelectron entanglement is necessarily present in both models, it does not lead to an additional photoionisation delay if the IR-driven coupling in the ion is switched off, i.e. if Path 3 in Fig. 4 is not included.

The second term of Eq. (5) can be expressed using the high-energy limit by means of one-photon observables. For a given relative orientation of the molecule and polarisation of the fields, and for a specific emission direction of the photoelectron, it is:

$$\tau_{ion} = \frac{1}{2k} \frac{\omega_{IR}}{\Delta - \omega_{IR}} \frac{\mathbf{D}_{BC} \cdot \hat{\boldsymbol{\epsilon}}_{IR}}{\hat{\boldsymbol{k}} \cdot \hat{\boldsymbol{\epsilon}}_{IR}} X(\gamma) \cos \arg\left(d_B^{(1)*} d_C^{(1)}\right), \qquad (6)$$

where $X(\gamma)$ monotonously decreases from 1 to 0 with purity $\gamma$ increasing from 1/2 to 1, acting as an overall scaling factor of the ion coupling delay (see SI 1.3 and Supplementary Fig. 10). The delay then depends on the modulus of the photoelectron momentum $k$, transition dipole $\mathbf{D}_{BC}$ between the coupled states $B^2\Sigma_u^+$ and $C^2\Sigma_g^+$, the phases of the XUV ionisation amplitudes $d_B^{(1)}$ and $d_C^{(1)}$, and the energy difference between the two states $\Delta = E_C - E_B$. Moreover, we note that the numerator and denominator of the expression depend on the relative angle between the direction of polarisation of the IR field (identified by the versor $\hat{\boldsymbol{\epsilon}}_{IR}$) and the transition dipole $\mathbf{D}_{BC}$, and the versor $\hat{\boldsymbol{\epsilon}}_{IR}$ and the direction of emission of the photoelectron $\hat{\boldsymbol{k}}$, respectively (see SI 1.3 for a detailed derivation and Supplementary Fig. 11 for a numerical illustration). The average purity $\gamma$ in the photon energy range between 22 eV and 40 eV is $0.53 \pm 0.02$ (see Supplementary Fig. 12).

In our model, photoelectrons/photoions resulting from XUV photoionisation are entangled even in the absence of the coupling between the ionic states. Therefore, the entangled nature of the photoelectron/photoion is unaffected by the presence or absence of this additional coupling. However, the coupling enables an additional attosecond time delay to be observed, depending on the entanglement between the photoelectron and photoion. We point out that ion-

photoelectron entanglement alone does not induce an additional delay. Nevertheless, entanglement is necessary in order to observe the additional delay originating from IR-induced ionic coupling.

We remark that the additional ion coupling time delay is not due to a Coulomb interaction between the photoelectron wave packet and the residual ion, but is a pure manifestation of the quantum interference enabled by the entangled nature of the photoelectron-photoion system. This follows from the fact that the ion-ion transition is solely mediated by the laser-ion interaction Hamiltonian $\mathbf{D} \cdot \mathbf{F}(t)$, which does not correspond to the interaction of the residual ion with the photoelectron but rather to field-driven dynamics in the residual ion (see also SI 1.3). Importantly, the delay induced on the photoelectron by the IR-induced coupling in the ion is independent from the ion-photoelectron distance.

The dependence of the photoionisation time delay on the angular integration region in the laboratory frame can be understood considering the vectorial character of the transition dipole $\mathbf{D}$ and the XUV-only photoelectron angular distribution from the $C^2\Sigma_g^+$ state. Due to the axial symmetry of the two orbitals (see Fig. 1b), the dipole $\mathbf{D}$ is parallel to the molecular axis. The ionic coupling delay is driven by the projection of the transition dipole along the field polarisation $\mathbf{D} \cdot \mathbf{F}(t)$. When the molecule is aligned along the polarisation direction of the IR field, the dipole transition between the $\Phi_B$ and $\Phi_C$ states is the strongest; after projection on the asymptotic field-free $\Phi_B$ state, the fraction of the dressed wave function initially in the $\Phi_C$ state on the total signal is the largest (see SI 1.3). When the molecular axis is perpendicular to the laser polarisation direction, the dipole transition vanishes.

The differential single-photon ionisation process for the $C^2\Sigma_g^+$ state is anisotropic as shown in Fig. 5, which presents the differential cross section at the photon energy of 29 eV by field polarised parallel with the molecular axis (Fig. 5a) and perpendicular to it (Fig. 5b). The laser polarisation and molecular axis direction are indicated by the red and blue arrows, respectively.

For photoelectrons emitted along the polarisation direction of the two fields (corresponding to the smallest integration angle presented in Fig. 3b), the major contributions come from molecules aligned parallel to the laser polarisation direction. Indeed, for molecules aligned parallel to the laser field, the differential cross section peaks along the polarisation direction (Fig. 5a) and reaches a much higher value with respect to the differential cross section along the same direction for molecules aligned perpendicular to the laser field (Fig. 5b). As a result, the largest fraction of photoelectrons originates from molecules oriented approximately parallel to the polarisation, maximising the effect of the dipole coupling in the residual ion.

When the photoelectron signal is integrated over the entire solid angle, the contribution of molecules not oriented along the laser polarisation direction increases (see, for example, the maxima of the differential cross section for molecules oriented perpendicular to the laser polarisation direction shown in Fig. 5b). On average, this reduces the effect of ionic coupling on the observed coupling delay. The anisotropy of the differential cross sections introduces a correlation between the molecular orientations that effectively contribute to the photoelectron spectrum and the polarisation direction of the fields, leading to an effect of ionic coupling that is stronger in the direction of the laser field. Therefore, the effect of the ionic coupling changes upon reducing the integration angle along the polarisation direction of the field, as demonstrated in Fig. 3b.

This conclusion is fully supported by the theoretical simulations with and without ionic coupling, as shown in Fig. 5c, d, respectively. It can be observed that without ionic coupling, the integration over the full solid angle leads to a small variation with respect to the parallel case, which disappears for high photon energies (Fig. 5d). The inclusion of the ionic coupling results in a significant reduction of the RABBIT delay[42], which is maximised when only the photoelectron emitted along the polarisation direction is considered (Fig. 5c). When

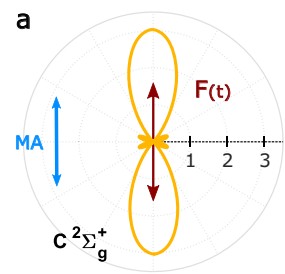

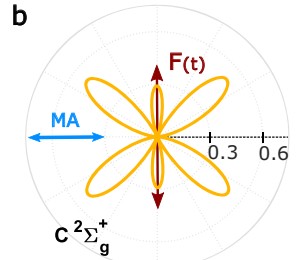

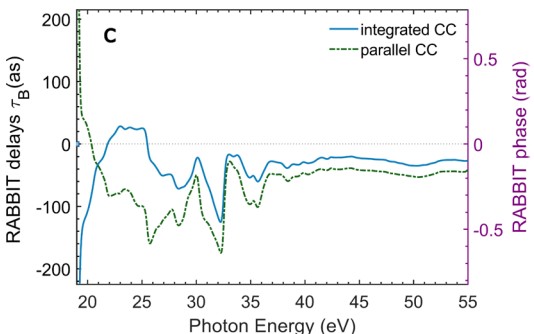

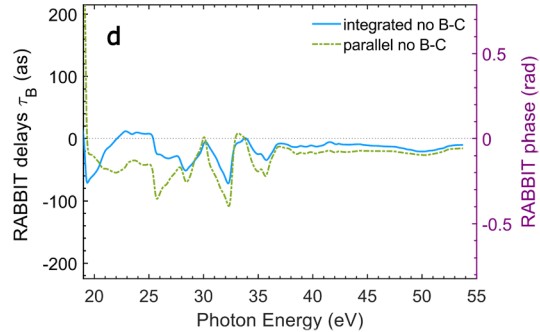

**Fig. 5 | Angular dependence of the effect of the coupling between the ionic states. a**, **b** Differential cross section for one-photon ionisation of $CO_2^+$ at photon energy 29 eV into the $C^2\Sigma_g^+$ state by field polarised parallel with the molecular axis (**a**) and perpendicular to it (**b**). The units are arbitrary, but consistent between the panels. The red and blue arrows indicate the laser polarisation direction and the molecular axis (MA), respectively. **c**, **d** RABBIT time delays $\tau_B$ for the $B^2\Sigma_u^+$ state integrating over the full solid angle (blue solid lines) or along the polarisation direction of the XUV and IR fields (green dash-dotted lines) with (**c**) and without (**d**) inclusion of the ionic coupling. For these simulations, the attochirp was set to zero.

integrating over the solid angle, the effect of the ionic coupling delay is strongly reduced (blue curve in Fig. 5c) and the RABBIT delay gets closer to the curve obtained by neglecting the effect of the ionic coupling (blue curve in Fig. 5d).

We have shown that attosecond time delays in photoionisation are sensitive to the dynamics occurring in the molecular ion due to the presence of the IR field; the external field can induce transitions not only in the continuum of states of the photoelectron wave packet, but also in the molecular ion. Our work indicates that RABBIT measurements can be contributed by more than two interfering paths when the effect of the IR field on the dynamics of the parent ion is included. The experimental data indicate that the attosecond time delays in photo-ionisation are a very sensitive probe of the interaction of IR fields with the molecular electronic structure of the cation. While in atoms the effect of the external infra-red field can often be neglected, this is generally not the case in molecules, where the rich electronic and vibrational energy structure can be responsible for couplings induced by the external radiation. Our experimental findings about the importance of laser-driven transitions in an ionic system in the near IR spectral range will also be relevant for the investigation of photoelectron-photoion entanglement using short wavelength radiation[58] produced at free-electron lasers generating multi-colour coherent harmonic radiation[59].

## Methods
### Experimental methods
Trains of attosecond pulses with photon energies up to $\approx 50$ eV were generated in krypton using 20 fs driving infra-red pulses centred at $\lambda \approx 1022$ nm at 50 kHz repetition rate. The time delay between the XUV radiation and the IR field was varied in a collinear geometry using a pair of drilled plates[48,49]. The two-colour field was focused at the interaction point of the reaction microscope using a toroidal mirror operating in one-to-one imaging at an angle of incidence of 84°. The gas target consisted of a mixture of randomly oriented $CO_2$

molecules and argon atoms. The typical count rate in the measurements was 5–6 kHz, and data were acquired for 76 h. The data discussed in the manuscript were integrated over all orientations of the molecules.

### Theoretical methods
For $CO_2$ we used the atom-centred Gaussian basis set cc-pVTZ, together with a set of uniformly distributed radial B-splines to represent the continuum, with partial wave expansion up to $\ell = 7$. The R-matrix radius was $R_a = 10$ atomic units. In case of the coupled "CC" model, one-electron orbitals were obtained using the complete active space self-consistent field (CASSCF) method with the core orbitals $1\sigma_g$, $2\sigma_g$ and $1\sigma_u$ frozen, and orbitals $3\sigma_g$, $4\sigma_g$, $5\sigma_g$, $1\pi_u$, $2\pi_u$, $2\sigma_u$, $3\sigma_u$ and $1\pi_g$ included in the active space. Three hundred states of the lowest energy obtained from multi-reference configuration interaction in the residual ion with the same active space as in CASSCF were then used in the close-coupling expansion. The two-photon amplitudes were calculated in the UKRmol+ package[60].

For Ar, we used the basis set cc-pVDZ with radial B-spline continuum orbitals, partial wave expansion up to $\ell = 4$ and R-matrix radius $R_a = 30$ atomic units. The "CHF-B" variant of the (non-relativistic) polarisation-consistent coupled Hartree-Fock model was used with Hartree-Fock orbitals of the neutral atom, with 5 frozen orbitals, 4 active valence orbitals and 9 virtual orbitals. The calculated excitation threshold of the 3s orbital with respect to 3p was manually adjusted to the experimental value to recover the correct position of the major $Ar^*(3s^13p^64p^1)$ and $Ar^*(3s^13p^64s^1)$ resonances.

The molecular orbitals presented in Fig. 1 were generated using IQMol (version 3.1.4; http://iqmol.org).

## Data availability
The data supporting the findings of this study are available at Ref. 61 (repository Zenodo under the https://doi.org/10.5281/zenodo.16913144).

## Code availability

Relevant codes presented in the manuscript are available on request from the authors.

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

## Acknowledgements

G.S. acknowledges financial support by FRIAS. G.S. and D.E. acknowledge financial support by the Deutsche Forschungsgemeinschaft project International Research Training Group (IRTG) CoCo 2079 and INST 39/1079 (High-Repetition-Rate Attosecond Source for Coincidence Spectroscopy), Priority Programme 1840 (QUTIF), grant 429805582 (Project SA 3470/4-1) and grant 546852490 (Project SA3470/13-1). G.S. and I.M. acknowledge financial support from the European Union's Horizon Europe research and innovation programme under the Marie Skłodowska-Curie grant agreement No 101168628 (project QU-ATTO). S.P. acknowledges financial support by Deutsche Forschungsgemeinschaft project PA 2691/3-1. I.M. and G.S. acknowledge financial support by the BMBF project 05K19VF1, the Deutsche Forschungsgemeinschaft project Research Training Group DynCAM (RTG 2717), and the Georg H. Endress Foundation. D.B. acknowledges support from the Swedish Research Council grant 2020-06384 and from the Knut and Alice Wallenberg Foundation through the Wallenberg Centre for Quantum Technology. This work has been supported by the Charles University Research Centre programme No. UNCE/24/SCI/016 and by the Ministry of Education, Youth and Sports of the Czech Republic through the e-INFRA CZ (ID:90254). J.B. and Z.M. acknowledge the support of the Czech Science Foundation (25-24428L). G.S., I.M. and D.B. acknowledge fruitful discussion with Andreas Buchleitner and Christoph Dittel.

## Author contributions

I.M., D.B., and D.E. developed the experimental setup. F.F. and L.P. collaborated in the development of the XUV spectrometer. C.D.S., T.P., R.M., and G.S. contributed to the development of the ReMi spectrometer. I.M., D.B., D.E., B.M., and B.S. performed the experiments. I.M. analysed the data. J.B. and Z.M. developed the numerical simulations for the calculation of the R-Matrix dipole element. S.P. contributed to the development of the theoretical model. I.M., D.B., J.B., Z.M., S.P., and G.S. interpreted the experimental data. G.S. conceived the idea of the experiment, supervised the work and wrote the manuscript, which was discussed and agreed by all coauthors.

## Funding

## Competing interests

The authors declare no competing interests.
