## [Transparent Peer Review file · Nature Communications]

Entanglement in photoionisation reveals the effect of ionic coupling in attosecond time delays

Corresponding Author: Professor Giuseppe Sansone

Version 0:

Reviewer comments:

Reviewer #1

(Remarks to the Author)

The manuscript "Entanglement in photoionisation reveals the effect of ionic coupling in attosecond time delays" reports on the experimental characterization of attosecond ionization delays from CO₂ molecules. The key experimental findings are the implications of internal dipole transitions in the CO₂⁺ ions and manifestations of photoelectron-photoion entanglement in the overall delay time response. The experimental method is state-of-the-art; it is a combination of 50-kHz ultrafast femtosecond lasers and a reaction microscope, and measurements were conducted over 76 hours of accumulation; a beautiful experimental work like this shows the exciting potential of the RABBIT technique to address important questions in molecular dynamics and photochemistry. On the theory side, the methods seem appropriate, and the results are supportive of the conclusions, although I would like to have some issues clarified. About the scientific impact, this work studies molecules not atoms, and investigates the role of the dipole transitions within the parent ions. This is an important question that needs to be revealed to extend the application of RABBIT to more complex chemical systems. The most interesting feature is the possible role of photoelectron-photoion entanglement. Entanglement is a hot topic in wide areas of natural sciences, and attosecond science is no exception. The attosecond community is still trying to understand how entanglement will be relevant in their experiments, and this work may become one of the very few first work to show experimental implications of entanglement in the attosecond experiments.

My detailed comments are as follows:

- (1) The abstract and the first three paragraphs of the introduction give me the impression that this is a RABBIT paper written for RABBIT experimentalists. It would be better if a little more emphasis was placed on the broader implications of the RABBIT measurements to the general physics community.
- (2) Fig. 3: without annotations within the figure, it is difficult to distinguish the curves and plots.
- (3) The simulations in Fig. 3 compare RABBIT delays with and without the B-C state coupling in the ion. I can understand that this establishes the importance of the B-C state coupling in the ion, but it is not clear to me why we need to invoke the ion-photoelectron entanglement (not just correlation). A more clear explanation of how the measurements will be different with and without the ion-photoelectron entanglement (while sustaining the B-C state coupling) is needed.
- (4) On a related note, I would like to know if the RABBIT measurements can tell us the degree of entanglement between the photoelectrons and photoions.
- (5) In lines 212-217, it is written that "the entangled nature of the photoelectron-photoion system ... is supported by the observation that the contribution does not depend on the distance between the photoelectron and photoion." I checked the main text and the supplementary material, but could not find such a result. Is the observation experimental, or theoretical? Which figure shows such data? This seems to be a critical piece of information to support the argument that the entanglement can be probed by the RABBIT technique.
- (6) A few typos in the manuscript:
A dot in the title seems unnecessary
Dots in the affiliations also seem unnecessary
Line 51: photon ionization -> photoionization

Reviewer #2

(Remarks to the Author)

In the work entitled "Entanglement in photoionisation reveals the effect of ionic coupling in attosecond time delays" by Makos

and co-workers, the authors study the effect of the laser-induced polarisation in parent ion during photoionisation on the photoelectrons time delay. The authors use their experimental observation on the CO₂ molecule to distinguish between the theoretical models describing photoionisation with and without the effect of the infrared field on the ion, and they are able to show the better agreement of the former with their experimental data.

The main result of this work is that attosecond time delays in photoionisation are sensitive to the dynamics occurring in the molecular ion due to the presence of the IR field. As the authors correctly conclude in their work, this is because the external field can induce transitions not only in the continuum of states of the photoelectron wave packet, but also in the molecular ion.

In general the results of this work are clearly presented throughout the manuscript, and are certainly of interest to the attosecond community.

However, there are a few issues that need to be resolved before I could recommend this work for publication:

C1. In their abstract the authors claim that "However, in general, due to the entanglement between the wave function of the emitted photoelectron and that of the parent ion, the dynamics occurring in the ion can affect the properties of the photoemitted electronic wave packet."

This is an incorrect and misleading statement. In a bipartite system, no Hermitian dynamics occurring only in one of the two sub-systems (in this case the parent ion) can affect the properties of the reduced density matrix of the other subsystem (in this case the entangled photoelectron). In other words, if measurements are performed on the photoelectron system only, Hermitian dynamics occurring only in the parent ion is not going to change their outcome. Entanglement results in the fact that different measurement results are obtained depending on how the full entangled state is measured, changes in the measured photoelectron properties will result only if the measurements on the photoelectron are correlated with measurements on the parent ion as well. Clearly, if dynamics occurs also in the photoelectron subsystem, i.e. if the whole system is subjected to the same dynamics, then this will affect the properties of each subsystem. But this is not what is implied by the authors statement. Therefore, this sentence needs to be either removed or edited accordingly.

C2. In their conclusions the authors claim that "Our results suggest that ion-photoelectron entanglement can be harnessed to probe the dynamics in the ion via the interferometric characterization of the photoelectron wave packet".

This statement is also very ambiguous and misleading. The same argument as in comment C1. applies to this sentence, which should therefore be removed. Characterization of the photoelectron wave packet, even full tomography of the reduced density matrix of the photoelectron system, does not give information on the dynamics in the parent ion. This is because the reduced density matrix of the photoelectron is obtained by tracing out the ionic degrees of freedom. In order to obtain information on the dynamics occurring in the ion, correlated measurements on both subsystem would need to be performed.

C3. In their abstract the authors claim that "The additional time delay stems from the entanglement between the photoion and the photoelectron created in the photoionisation process".

The authors correctly state that resonant IR-driven dipole transitions between states of the parent ion affects the photoelectrons time delay. In this mechanism, the photoelectron acts as a spectator and does not change its state during the rearrangement of the electronic cloud within the ionic system. A quantum amplitude referred to a fixed photoelectron state is transferred between the ionic states involved in the dipole transition.

Intuitively, the photoelectrons time delay are in general affected because the photoelectron does not see a stable electronic cloud within the ionic system, but rather one that changes during the ionic transitions. This effect does not directly require entanglement, therefore attributing a general and direct role to entanglement is misleading. As an example, assume that the XUV populates ionic state A with photoelectron i , as well as ionic state B with photoelectron i , and then consider the following two quantum paths: in the first one, the IR does not cause any transition, we still have ionic state A with photoelectron i ; in the second one, the IR only induces transition in the ionic system (from B to A). The interference of these two paths will also influence the time delay, while there is no entanglement at any stage of these quantum paths.

Moreover, in the present manuscript there is no quantification of the role of entanglement, i.e. how a stronger/weaker entanglement in the photoelectron-ion state potentially leads to a stronger/weaker change in the photoelectrons time-delay. In general, it is more appropriate to say that the multi-channel nature of the photoionization plays a direct role, rather than entanglement (although in the past few years it has become fancy to emphasize entanglement's role even when not needed).

Therefore, I recommend the authors to edit the above sentence (The additional time delay can be affected by the entanglement between the photoion and the photoelectron created in the photoionisation process" is only a suggestion...).

C4. I also strongly recommend the authors to drop the word entanglement from the title of the manuscript, as the current title misleadingly attributes too much direct importance to entanglement.

C5. The reference list is quite short, and in particular it misses some important references that are very relevant to this work such as:

In [Phys. Chem. Chem. Phys. 20 , 8311 (2018)] the role of laser-driven dipole transitions between different molecular ionic states on the photoionization electron dynamics was studied and uncovered in the context of multichannel high-order harmonic generation in the CO₂ molecule.

In [Journal of Chemical Theory and Computation 14 , 4991 (2018)] the electron rearrangement within the parent ion, caused by laser-driven dipole transitions between the different ionic states, was studied in relation to the onset of quantum coherence within the CO₂ cation upon strong-field ionization. Resonant or nearly-resonant transitions between states of the parent ion that are coupled to each other by an electric dipole matrix element was also identified as one of the main mechanisms of ionic coherence formation (and conversely formation of entanglement between parent ion and photoelectron) in photoionization in [Phys. Chem. Chem. Phys. 24, 19673 (2022)].

In [New Journal of Physics 21, 113036 (2019)] the consequences of photoelectron – parent ion entanglement in interferometric experiments observations was addressed. In particular it was shown how the existence (due to entanglement) of multiple quantum interference channels involving incoherently populated ionic states can lead to a suppression of quantum-path interference in some photoelectron observables.

I recommend the authors to update their reference list and include the aforementioned works.

Moreover, some of the most recent and important works on quantum entanglement in photoionization are also missing, in inverse chronological order:

- H. Laurell, et al. Measuring the quantum state of photoelectrons, Nat. Photon. (2025). <https://doi.org/10.1038/s41566-024-01607-8>
 - M Ruberti, V Averbukh, F Mintert, Bell test of quantum entanglement in attosecond photoionization, Physical Review X 14, 041042 (2024).
 - F. Shobeiry, P. Fross, H. Srinivas, T. Pfeifer, R. Moshhammer, and A. Harth, Emission control of entangled electrons in photoionisation of a hydrogen molecule, Sci. Rep. 14, 19630 (2024).
 - S. Eckart, et al. Ultrafast preparation and detection of entangled atoms, Sci. Adv. 9, eabq8227 (2023).
 - A. S. Maxwell, L. B. Madsen, and M. Lewenstein, Entanglement of orbital angular momentum in nonsequential double ionization, Nat. Commun. 13, 4706 (2022).
 - H. Laurell, et al. Continuous-variable quantum state tomography of photoelectrons, Physical Review Research 4, 033220 (2022).
 - C. Bourassin-Bouchet, et al. Quantifying decoherence in attosecond metrology, Phys. Rev. X 10, 031048 (2020).
- I recommend the authors to update their reference list and also include these works.

To conclude, I would recommend this work for publication in Nature Communications only after these problems (C1-C5) are fully resolved by the authors.

Reviewer #3

(Remarks to the Author)

Makos et al report on a RABBIT-type experiment on CO₂ that shows attosecond delays that are traced back to the polarization of the produced CO₂⁺ ion. By comparison with theory, the authors conclude that the time delay is due to the entanglement of photoion and -electron.

The experiment is very sophisticated indeed: The RABBIT technique is employed in a reaction microscope, allowing coincidence detection of the photoions and photoelectrons. In particular, this device enables to measure RABBIT traces of Argon and CO₂ simultaneously, such that the attosecond delays of the CO₂'s photoelectrons can be measured relative to the Argon photoelectrons in one and the same focus of one and the same laser pulse. Very elegant!

The evidence of ionization delays caused by entanglement / polarization of the photoion rests on the two data points at 28.8 and 31.2 eV in Fig. 3. They lie – within error bars – on the black curve representing a model that includes polarization. However, the other two curves are within not more than one error bar. Assuming that the error bars represent the usual 1-sigma confidence level, there is a probability of more than 10% that the effect is of statistical nature.

The behavior presented in Fig 3b could provide stronger support to the claims put forward as experimental data and theory show the same trend in dependence of the detection angle – at least for the data points at 26.4, 28.8, and 31.2 eV, namely a shift to more negative delays for decreasing detection angle. However, for lower energies this is not the case. Why should one accept the behavior at >26 eV as a proof, if the model does not agree below 26 eV, at 24 eV even pretty strongly?

What we do not understand is why overlapping detection angles were used. Why 0 - 20°, 0 - 50°, and 0 - 90° instead of 0 - 20°, 20° - 50°, and 50° - 90°. Obviously, the overlapping detection angles reduce the magnitude of the effect. Why would one present a deliberately reduced effect? What is the reason for this choice?

Our critique so far mainly concerns statistical uncertainties. (We believe there should be 3-sigma confidence, at least, to make a claim). Quite obviously, there are also systematic factors – in the experiment as well as in the theoretical model. Our conclusion is that there are indications that the polarization of the CO₂ molecular ion causes a delay on photoelectron emission (which is hardly surprising, but potentially interesting), but there is not yet evidence, not to speak of proof.

We also have a number of remarks that may be of more technical nature:

- The spectrum Fig 1c shows features not explained in the text, in particular the switching of the peaks from the green to the red line. What is the explanation and what are the implications for the experiment and its interpretation.
- The separation of the lines in Fig 1c does not match the photon energy, it is not even constant.
- The text states that "the experimental data clearly show a linear variation". We clearly see a curvature. Is this reproduced by theory?

Reviewer #4

(Remarks to the Author)

I co-reviewed this manuscript with one of the reviewers who provided the listed reports. This is part of the Nature

Communications initiative to facilitate training in peer review and to provide appropriate recognition for Early Career Researchers who co-review manuscripts.

Version 1:

Reviewer comments:

Reviewer #1

(Remarks to the Author)

I was the Reviewer #1 for the manuscript "Entanglement in photoionisation reveals the effect of ionic coupling in attosecond time delays." This is my second review. My first review contained six comments, and I appreciate the efforts by the authors to address them. Here are my responses to the revision by the authors.

(1) The revised introduction gives better explanations of the RABBIT technique. I appreciate the efforts by the authors. There is no more question from my end.

(2) The added annotations in Figure 3 are helpful, but the figure still looks congested. In my personal opinion, plotting six curves in a single figure (Fig. 3(b)) is not a good idea. There is no more question from my end, though.

(3) My comment #3 was about the distinction of correlation and entanglement. The response by the authors is difficult for me to follow.

(3-1) It remains unclear to me, in the model, how we can compare the classical ion-electron pairs and the entangled ion-electron pairs. If with/without BC coupling is taken equivalent to entangled/classical pairs, that I think should be clearly stated so.

(3-2) I understand that the additional photoionization delay is the key information about the entanglement, and that corresponds to the deviation between the black curve (th. CC) and the blue curve (th. no B-C) in Fig. 3(a). I need to point out that the deviation and the experimental error bars are about the same amplitude. I would like to know how much confidence the authors have about the importance of the B-C coupling when compared to the experimental data.

(4) The new estimate of the purity is useful and I appreciate the efforts by the authors to include this number.

The lines 274-275 "The purity ... is about 0.5-0.6" make a strange phrase. There should be just one number and a standard deviation, not a range.

(5) The newly added text is helpful. My original question arose because "the observation that the contribution does not depend on the distance between the photoelectron and photoion" sounded experimental, and I wanted to see such data. But now it seems that this is all within the theoretical model and there is no experimental evidence. There is no more question from my end.

(6) The typos have been nicely fixed. There is no more question from my end.

Overall, I think that the quality of experimental and theoretical data in the manuscript is about the average of articles in Nature Communications, and this can be a good stimulating paper for the community. My reservation is that the level of confidence in overall arguments needs to be carefully adjusted, especially around the role of entanglement. For example, changing the title (also suggested by the reviewer #2) seems appropriate. In my personal view, this paper can be one of the early works that suggest or try to address the potential implication of attosecond entanglement. I envision that more future works will follow with improved experimental design and statistics to establish the experimental manifestation of entanglement in attosecond experiments.

Reviewer #2

(Remarks to the Author)

The authors have fully addressed all my previous comments in a detailed and convincing way.

I also believe that, in the revised manuscript, the discussion on the connection between entanglement and photoionization time delay is certainly more complete and rigorous than it was before.

Therefore, I am happy to recommend the work "Entanglement in photoionisation reveals the effect of ionic coupling in attosecond time delays" for publication in Nature Communications in the current form.

Reviewer #3

(Remarks to the Author)

The central critique of our first report have been doubts about the significance of the results in the sense of statistical and systematic errors. We had asked what the confidence level is, particularly for the data points at 28.2eV and 31eV in Fig 3a. This question, which in our opinion is a crucial one in physics, is not addressed in the authors' answer, not to say that it is being evaded.

The authors claim that the strong deviation of the data point at 24eV is due to "limitations of the theoretical model in accurately describing the electronic states within this region", where "this" refers to the vicinity of the ionization threshold. However, the C state has a binding energy of 19.4eV, almost 5eV less. In addition, the points closer to the ionization threshold fit better. Moreover, the paper claims that the model "simulates the correlations with high accuracy" (l. 174). What is to be understood by „high accuracy“?

In our opinion, two things are mandatory before publication can be considered

- It MUST be shown that the deviation of the data points at 28.2eV and 31eV from the non-complete models is significant

with at least 3 sigma (or another criterion that is similarly convincing)

- It MUST be explained convincingly why the model(s) deviate strongly from the experiment at 24eV and why the models are nevertheless reliable at the other energies.

Finally, we (and reviewer 1) ask ourselves whether framing the work in terms of entanglement helps in elucidating anything. After all, it has always been obvious (or, at least, should have been obvious) that the fragments of photoionization and -dissociation are entangled. Is it just new language (new in the attosecond community) or does it provide insight that may help to advance the field?

We are also a bit puzzled about the role of the IR field: It affects the properties of the photoelectron and, at the same time, is used to measure the delay. It seems that both roles are, well, entangled. A natural specific question in this respect would be if the measured delay depends on the IR field strength?

Reviewer #4

(Remarks to the Author)

Version 2:

Reviewer comments:

Reviewer #1

(Remarks to the Author)

The response letter contained some useful information, but it was not added to the main manuscript. Adding the following two sentences will help make the work accessible to the broad readership of Nat Commun:

“In our model, photoelectrons/photoions resulting from XUV photoionisation are entangled even in the absence of BC coupling. Therefore, the entangled nature of the photoelectron/photoion is unaffected by the presence or absence of BC coupling. However, BC coupling enables an additional attosecond time delay to be observed, depending on the entanglement between the photoelectron and photoion.”

“We point out that ion-photoelectron entanglement alone does not induce an additional delay. However, entanglement is necessary in order to induce an additional delay originating from IR-induced BC coupling.”

Reviewer #3

(Remarks to the Author)

The authors have carefully considered our concerns. Their analysis indicates a sufficient confidence of the validity of their conclusions.

As a remark: It is true that it is general practice, not only in attoscience, to indicate 1-sigma confidence intervals on data points – like in, e.g., [2] Isinger et al. To illustrate our point: If there had been only one gray diamond data point in their Fig 1A, the case would have been weak. However, there are five such points, all deviating more than 1 sigma in the same direction.

In the present case, there 2 times 3 data points, however they use partially the same events. On the other hand, two independent points with a 1% probability that they are compatible with the negative hypothesis are satisfactory.

Reviewer #4

(Remarks to the Author)

Point-by-point response for the manuscript:

Entanglement in photoionisation reveals the effect of ionic coupling in attosecond time delays
Nature Communications manuscript NCOMMS-24-81266-T

We would like to thank the reviewers for their remarks. According to their indications and suggestions, we have prepared a revised version of the manuscript. Hereafter, we include our point-by-point response to their comments. We have adopted the following color-code to simplify the presentation of our answers and to illustrate the changes introduced in the revised manuscript:

- **In black**, we report the original comment/remark of the reviewer:
- **In red**, we highlight our answers.
- **In blue**, we indicate the main changes introduced in the revised version of the manuscript.

In the resubmission, we provide two versions of the main manuscript and supplementary information. In the first, we have highlighted all major changes in the revised version of our paper, while the second is a clean version of the revised work.

The pages and lines used to indicate the location of the changes discussed in this point-by-point response refer to the version of the manuscript in which the major changes are highlighted.

Reviewer #1 (Remarks to the Author):

The manuscript “Entanglement in photoionisation reveals the effect of ionic coupling in attosecond time delays” reports on the experimental characterization of attosecond ionization delays from CO₂ molecules. The key experimental findings are the implications of internal dipole transitions in the CO₂⁺ ions and manifestations of photoelectron-photoion entanglement in the overall delay time response. The experimental method is state-of-the-art; it is a combination of 50-kHz ultrafast femtosecond lasers and a reaction microscope, and measurements were conducted over 76 hours of accumulation; a beautiful experimental work like this shows the exciting potential of the RABBIT technique to address important questions in molecular dynamics and photochemistry. On the theory side, the methods seem appropriate, and the results are supportive of the conclusions, although I would like to have some issues clarified. About the scientific impact, this work studies molecules not atoms, and investigates the role of the dipole transitions within the parent ions. This is an important question that needs to be revealed to extend the application of RABBIT to more complex chemical systems. The most interesting feature is the possible role of photoelectron-photoion entanglement. Entanglement is a hot topic in wide areas of natural sciences, and attosecond science is no exception. The attosecond community is still trying to understand how entanglement will be relevant in their experiments, and this work may become one of the very few first work to show experimental implications of entanglement in the attosecond experiments.

We thank the reviewer for her/his positive assessment of the impact of our work.

My detailed comments are as follows:

(1) The abstract and the first three paragraphs of the introduction give me the impression that this is a RABBIT paper written for RABBIT experimentalists. It would be better if a little more emphasis was placed on the broader implications of the RABBIT measurements to the general physics community.

We have introduced the following paragraphs to remark the sensitivity of the RABBIT approach to the electronic structure of the target system (page 3 lines 52-58):

This technique is extremely sensitive to the electronic structure of the target systems and in particular to the presence of resonances such as autoionising resonances,^{13, 14} shape resonances^{15, 16}, and shake-up states¹⁶. The interferometric nature of the RABBIT technique provides a high temporal resolution that has been used to resolve in time the formation of Fano profiles¹⁷ and to characterise the role of Cooper minima in attosecond time delays¹⁸. As such, this approach can be used to benchmark theoretical models describing the ultrafast response of correlated multielectron systems to an external light field.

(2) Fig. 3: without annotations within the figure, it is difficult to distinguish the curves and plots. We have introduced annotations in the revised version of the figure to improve its readability.

(3) The simulations in Fig. 3 compare RABBIT delays with and without the B-C state coupling in the ion. I can understand that this establishes the importance of the B-C state coupling in the ion, but it is not clear to me why we need to invoke the ion-photoelectron entanglement (not just correlation). A more clear explanation of how the measurements will be different with and without the ion-photoelectron entanglement (while sustaining the B-C state coupling) is needed.

The electronic correlation, mediated by Coulomb interaction, affects the photoelectron wavefunctions and both our CC models do include significant correlation. This type of field-free correlation couples the B and C states via continuum (long-range dipolar interaction). It is naturally included in all our CC models. When the field is included in the perturbative development, the transitions between these correlated states are also driven by the IR field and the corresponding dipole transitions governed by operators $\mathbf{r} \cdot \mathbf{F}$ (for field-photoelectron interaction) and $\mathbf{D} \cdot \mathbf{F}$ (for field-residual ion interaction causing the B-C transition). Setting \mathbf{D} to zero effectively disables this dipole transition, including the IR-field-driven dynamics in the residual ion discussed in the manuscript. Finally, the ion-photoelectron entanglement is necessarily present due to the different channel energies of the photoelectrons coupled to the ionic wavefunctions and cannot be switched off. To address the reviewers' comment, we have extended the introduction of the theoretical models in the following way (page 9 lines 174-178):

Both CC models simulate electronic correlation to a high accuracy⁵⁴ in the initial, intermediate and the final state. The difference between the two models is in the transition from the XUV-induced intermediate continuum state to the final continuum state, which can be in principle mediated by IR-photoelectron interaction as well as by IR-ion interaction. The second model removes the possibility of the IR-induced transition between $B \ ^2\Sigma_u^+$ and $C \ ^2\Sigma_g^+$ states.

In the discussion we now explicitly comment on the role of the entanglement in the two CC models (page 13 lines 260-262):

While ion-photoelectron entanglement is necessarily present in both models, it does not lead to an additional photionisation delay if the IR-driven coupling in the ion is switched off, i.e. if Path 3 in Fig. 4 is not included.

Moreover, we have added a new equation (Eq. (7) page 13 line 265) presenting the connection between the degree of entanglement (quantified through the purity γ) and the additional attosecond time delay (τ_{ion})

(4) On a related note, I would like to know if the RABBIT measurements can tell us the degree of entanglement between the photoelectrons and photoions.

In the revised manuscript we included a new equation (Eq. (7)) expressing the dependence of ionic time delay τ_{ion} from the the purity γ of the reduced density matrix of the residual-ion two-level subsystem conditioned on the energy absorbed from the harmonic $2q+1$. In the description around this new equation (page 13), we show that this quantity is directly linked to the additional ion-ion coupling delay discussed in the manuscript. Moreover, to further clarify the connection between entanglement (purity) and attosecond time delays we have revised the section „Discussion“ in the main manuscript (pages 10-15) and we have added two new sections („Reduced density matrix“ and „Purity and coupling delay“) and two new figures (Extended Data Fig. 8 and Extended Data Fig. 9) in the supplementary information (pages 15-22).

The purity is, asymptotically, a function of XUV-only cross sections and, in principle, can be calculated therefrom.

In the molecular frame the direct link between the entanglement and the coupling delay (mediated by ionic IR transitions) is most apparent: for zero entanglement the coupling delay is zero, while the

coupling delay is maximized for perfect entanglement. Importantly, the degree of entanglement depends on the direction of the ejected photoelectron. Therefore in the laboratory frame the link between the entanglement and the coupling delay is not as simple due to the orientational averaging but is clearly present.

In the revised version of the manuscript, we also add a Figure showing the evolution of purity as a function of the photon energy (see Extended Data Fig.9 page 22).

Moreover, we have added the following sentence in the main manuscript (page 14 lines 274-275)

The purity γ in the photon energy range between 22 and 40 eV is about 0.5-0.6 (see Extended Data Fig. 10)

(5) In lines 212-217, it is written that “the entangled nature of the photoelectron-photoion system ... is supported by the observation that the contribution does not depend on the distance between the photoelectron and photoion.” I checked the main text and the supplementary material, but could not find such a result. Is the observation experimental, or theoretical? Which figure shows such data? This seems to be a critical piece of information to support the argument that the entanglement can be probed by the RABBIT technique.

We thank the reviewer for pointing out this unclear sentence, which has been removed from the revised manuscript. This point is now addressed as follows (pages 15 lines 281-285):

This follows from the fact that ion-ion transition is solely mediated by the laser-ion interaction Hamiltonian $D \cdot F(t)$, which does not correspond to interaction of the residual ion with the photoelectron but rather to field-driven dynamics in the residual ion, (see also SM 1.3). Importantly, delay induced on the photoelectron by the IR-induced coupling in the ion is independent from the ion-photoelectron distance.

(6) A few typos in the manuscript:

A dot in the title seems unnecessary

Dots in the affiliations also seem unnecessary

Line 51: photon ionization -> photoionization

We have corrected the typos in the revised version of the manuscript.

Reviewer #2 (Remarks to the Author):

In the work entitled “Entanglement in photoionisation reveals the effect of ionic coupling in attosecond time delays” by Makos and co-workers, the authors study the effect of the laser-induced polarisation in parent ion during photoionisation on the photoelectrons time delay. The authors use their experimental observation on the CO₂ molecule to distinguish between the theoretical models describing photoionisation with and without the effect of the infrared field on the ion, and they are able show the better agreement of the former with their experimental data.

The main result of this work is that attosecond time delays in photoionisation are sensitive to the dynamics occurring in the molecular ion due to the presence of the IR field. As the authors correctly conclude in their work, this is because the external field can induce transitions not only in the continuum of states of the photoelectron wave packet, but also in the molecular ion.

In general the results of this work are clearly presented throughout the manuscript, and are certainly of interest to the attosecond community.

However, there are a few issues that need to be resolved before I could recommend this work for publication:

C1. In their abstract the authors claim that “However, in general, due to the entanglement between the wave function of the emitted photoelectron and that of the parent ion, the dynamics occurring in the ion can affect the properties of the photoemitted electronic wave packet.”

This is an incorrect and misleading statement. In a bipartite system, no Hermitian dynamics occurring only in one of the two sub-systems (in this case the parent ion) can affect the properties of the reduced density matrix of the other subsystem (in this case the entangled photoelectron). In other words, if measurements are performed on the photoelectron system only, Hermitian dynamics occurring only in the parent ion is not going to change their outcome. Entanglement results in the fact that different measurement results are obtained depending on how the full entangled state is measured, changes in the measured photoelectron properties will result only if the measurements on the photoelectron are correlated with measurements on the parent ion as well. Clearly, if dynamics occurs also in the photoelectron subsystem, i.e. if the whole system is subjected to the same dynamics, then this will affect the properties of each subsystem. But this is not what is implied by the authors statement. Therefore, this sentence needs to be either removed or edited accordingly.

We thank the reviewer for pointing out this inaccuracy. We have reworded the sentence as follows (page 2 lines 28-33):

However, in general, due to the entanglement between the wave function of the emitted photoelectron and that of the parent ion, the dynamics driven by the infrared field in the photoion can affect the properties of the photoemitted electronic wave packet, when the measurement protocol corresponds to the projection of the total time-dependent wave function onto a specific final state of the bipartite system.

Moreover, we have added the following text to point out the fact that we are performing coincidence measurements of photoelectrons and photoions (pag. 13 lines 250-255):

We note that photoionisation from the $B \ ^2\Sigma_u^+$ state does not lead to molecular dissociation⁵⁶, while the $C \ ^2\Sigma_g^+$ state completely predissociates⁵⁷. The combination of these two observations ensures that the coincidence measurements of the photoion CO₂⁺ with the photoelectrons in the energy range around the expected values of the sidebands associated with the $B \ ^2\Sigma_u^+$ state (see Fig. 1c) correspond to a projection of the total wave function of the system Ψ onto the final state of the bipartite system Ψ_{B2q} .

C2. In their conclusions the authors claim that “Our results suggest that ion-photoelectron entanglement can be harnessed to probe the dynamics in the ion via the interferometric characterization of the photoelectron wave packet”.

This statement is also very ambiguous and misleading. The same argument as in comment C1. applies to this sentence, which should therefore be removed. Characterization of the photoelectron wave packet, even full tomography of the reduced density matrix of the photoelectron system, does not give information on the dynamics in the parent ion. This is because the reduced density matrix of the photoelectron is obtained by tracing out the ionic degrees of freedom. In order to obtain information on the dynamics occurring in the ion, correlated measurements on both subsystem would need to be performed.

We thank again the reviewer for pointing out this misleading formulation. We have removed this sentence in the revised version of the manuscript.

C3. In their abstract the authors claim that “The additional time delay stems from the entanglement between the photoion and the photoelectron created in the photoionisation process”.

The authors correctly state that resonant IR-driven dipole transitions between states of the parent ion affects the photoelectrons time delay. In this mechanism, the photoelectron acts as a spectator and does not change its state during the rearrangement of the electronic cloud within the ionic system. A quantum amplitude referred to a fixed photoelectron state is transferred between the ionic states involved in the dipole transition.

Intuitively, the photoelectrons time delay are in general affected because the photoelectron does not see a stable electronic cloud within the ionic system, but rather one that changes during the ionic transitions. This effect does not directly require entanglement, therefore attributing a general and direct role to entanglement is misleading.

As an example, assume that the XUV populates ionic state $|A\rangle$ with photoelectron $|i\rangle$, as well as ionic state $|B\rangle$ with photoelectron $|i\rangle$, and then consider the following two quantum paths: in the first one, the IR does not cause any transition, we still have ionic state $|A\rangle$ with photoelectron $|i\rangle$; in the second one, the IR only induces transition in the ionic system (from $|B\rangle$ to $|A\rangle$). The interference of these two paths will also influence the time delay, while there is no entanglement at any stage of these quantum paths.

We thank the reviewer for sharing his/her views on the origin of the observed effect. However, we do not agree with his/her interpretation of the physical origin of the effect. The statement that the photoelectron does not see a stable electronic cloud within the ionic system, but rather one that changes during the ionic transitions, implies that the photoelectron leaving the molecular system experiences the changing electronic cloud in the cationic system through an interaction (e.g. Coulomb interaction). However, as noted in the revised version of the manuscript, the interaction does not depend on the ion-photoelectron distance and can occur at a much later time (i.e. when the photoelectron has already left the vicinity of the parent cation), leading to the same effect (introduction of an additional time delay). This observation indicates that the effect is not mediated by a Coulomb interaction, but is the result of the entanglement of the photoelectron-photoion system.

Moreover, in the present manuscript there is no quantification of the role of entanglement, i.e. how a stronger/weaker entanglement in the photoelectron-ion state potentially leads to a stronger/weaker change in the photoelectrons time-delay.

We thank again the reviewer for this remark. In the revised version of the manuscript, we have deepened the discussion about the connection between entanglement and photoionisation time delay (see pages 12,13 and 14), by introducing the degree of entanglement through the purity γ and showing how this parameter is related to the observed shift in time delay τ_{ion} .

Indeed in Eq. (7) we present an explicit formula, at least for a situation with resolved emission direction and molecular orientation, which highlights the clear link between the additional time delay and the purity.

In general, it is more appropriate to say that the multi-channel nature of the photoionization plays a direct role, rather than entanglement (although in the past few years it has become fancy to emphasize entanglement's role even when not needed).

Therefore, I recommend the authors to edit the above sentence (The additional time delay can be affected by the entanglement between the photoion and the photoelectron created in the photoionisation process” is only a suggestion...).

We thank the reviewer for this comment, but we prefer to keep the original wording in order to clarify the connection with the entanglement of the photoelectron-photoion system (see also next point).

C4. I also strongly recommend the authors to drop the word entanglement from the title of the manuscript, as the current title misleadingly attributes too much direct importance to entanglement. As pointed out by the reviewer, and given the number of papers on entanglement in attosecond and strong field science, we feel it is important to highlight the connection of our work to this main topic in the ultrafast community.

Moreover, in the revised version of the manuscript we show the connection between the observed additional delay τ_{ion} and the purity γ (see Eq. (7) and new sections in the Supplementary Information). For this reason, we prefer to keep the original title of our submission.

C5. The reference list is quite short, and in particular it misses some important references that are very relevant to this work such as:

In [Phys. Chem. Chem. Phys. 20 , 8311 (2018)] the role of laser-driven dipole transitions between different molecular ionic states on the photoionization electron dynamics was studied and uncovered in the context of multichannel high-order harmonic generation in the CO₂ molecule.

In [Journal of Chemical Theory and Computation 14 , 4991 (2018)] the electron rearrangement within the parent ion, caused by laser-driven dipole transitions between the different ionic states, was studied in relation to the onset of quantum coherence within the CO₂ cation upon strong-field ionization. Resonant or nearly-resonant transitions between states of the parent ion that are coupled to each other by an electric dipole matrix element was also identified as one of the main mechanisms of ionic coherence formation (and conversely formation of entanglement between parent ion and photoelectron) in photoionization in [Phys. Chem. Chem. Phys. 24, 19673 (2022)].

In [New Journal of Physics 21, 113036 (2019)] the consequences of photoelectron – parent ion entanglement in interferometric experiments observations was addressed. In particular it was shown how the existence (due to entanglement) of multiple quantum interference channels involving incoherently populated ionic states can lead to a suppression of quantum-path interference in some photoelectron observables.

I recommend the authors to update their reference list and include the aforementioned works.

We have included them in the reference list of the new version of the manuscript by revising the Introduction (page 4) and Conclusions (page 19) sections of the manuscript.

Moreover, some of the most recent and important works on quantum entanglement in photoionization are also missing, in inverse chronological order:

- H. Laurell, et al. Measuring the quantum state of photoelectrons, Nat. Photon. (2025). <https://doi.org/10.1038/s41566-024-01607-8>

- M Ruberti, V Averbukh, F Mintert, Bell test of quantum entanglement in attosecond photoionization, Physical Review X 14, 041042 (2024).

- F. Shobeiry, P. Fross, H. Srinivas, T. Pfeifer, R. Moshhammer, and A. Harth, Emission control of entangled electrons in photoionisation of a hydrogen molecule, Sci. Rep. 14, 19630 (2024).

- S. Eckart, et al. Ultrafast preparation and detection of entangled atoms, Sci. Adv. 9, eabq8227 (2023).

- A. S. Maxwell, L. B. Madsen, and M. Lewenstein, Entanglement of orbital angular momentum in nonsequential double ionization, Nat. Commun. 13, 4706 (2022).

- H. Laurell, et al. Continuous-variable quantum state tomography of photoelectrons, Physical Review Research 4, 033220 (2022).

- C. Bourassin-Bouchet, et al. Quantifying decoherence in attosecond metrology, Phys. Rev. X 10, 031048 (2020).

I recommend the authors to update their reference list and also include these works.

We appreciate the reviewer's suggestions for additional references and have incorporated the suggested works into our reference list.

To conclude, I would recommend this work for publication in Nature Communications only after these problems (C1-C5) are fully resolved by the authors.

Reviewer #3 (Remarks to the Author):

Makos et al report on a RABBIT-type experiment on CO₂ that shows attosecond delays that are traced back to the polarization of the produced CO₂⁺ ion. By comparison with theory, the authors conclude that the time delay is due to the entanglement of photoion and -electron.

The experiment is very sophisticated indeed: The RABBIT technique is employed in a reaction microscope, allowing coincidence detection of the photoions and photoelectrons. In particular, this device enables to measure RABBIT traces of Argon and CO₂ simultaneously, such that the attosecond delays of the CO₂'s photoelectrons can be measured relative to the Argon photoelectrons in one and the same focus of one and the same laser pulse. Very elegant!

We thank the reviewer for this positive comment about our approach.

The evidence of ionization delays caused by entanglement / polarization of the photoion rests on the two data points at 28.8 and 31.2 eV in Fig. 3. They lie – within error bars – on the black curve representing a model that includes polarization. However, the other two curves are within not more than one error bar. Assuming that the error bars represent the usual 1-sigma confidence level, there is a probability of more than 10% that the effect is of statistical nature.

The behavior presented in Fig 3b could provide stronger support to the claims put forward as experimental data and theory show the same trend in dependence of the detection angle – at least for the data points at 26.4, 28.8, and 31.2 eV, namely a shift to more negative delays for decreasing detection angle. However, for lower energies this is not the case. Why should one accept the behavior at >26 eV as a proof, if the model does not agree below 26 eV, at 24 eV even pretty strongly?

The discrepancy observed in the vicinity of the ionisation threshold can be attributed to the limitations of the theoretical model in accurately describing the electronic states within this region. Consequently, the observed discrepancy between experimental data and theoretical simulations in this photoelectron kinetic energy region is not entirely unexpected.

What we do not understand is why overlapping detection angles were used. Why 0 - 20°, 0 - 50°, and 0 - 90° instead of 0 - 20°, 20° - 50°, and 50° - 90°. Obviously, the overlapping detection angles reduce the magnitude of the effect. Why would one present a deliberately reduced effect? What is the reason for this choice?

The observed increase of the effect with decreasing integration angle is the combination of two effects:

- 1) the photoelectron angular distributions in the laboratory frame for different molecular orientation directions (the photoelectron angular distribution is stronger for molecules oriented parallel to the laser radiation and peaks in this direction).
- 2) The strength of coupling in the ionic states as a function of molecular orientation (the effect is maximised for molecules aligned parallel to the laser field). This point is also highlighted in the new Eq. (7).

By considering a smaller integration angle along the polarisation direction, we essentially increase the contribution of molecules aligned parallel to the laser radiation (due to point 1); these molecules show the largest ionic coupling (due to point 2). The combination of these two effects explains why the delay shift is larger for small integration angles along the common polarisation direction of the XUV and IR fields.

As the reviewer correctly points out, the effect diminishes with increasing integration angles (because the contribution of other molecular orientations, for which the coupling is reduced, increases), but we do not think that this effect is obvious; rather it further supports the conclusion of the coupling of the ionic system as origin of the attosecond time delay.

For the sake of completeness, we present below the evolution of the RABBIT delays for neighbouring (0° -20°, 20° -40° and 40° -90°) integration intervals, as suggested by the reviewer.

Figure 1: Evolution of the RABBIT delay τ_B for different angular integration intervals along the laser polarisation.

Although, as expected, a trend can be observed in the experimental data for neighbouring integration intervals, the interpretation of the effect is not as straightforward as that currently presented in the manuscript. In fact, in this representation it is not immediately obvious how the change in the integration range is related to the shift in the attosecond time delays. Obviously, the effect is still maximum for the first point (0° -20°), but the evolution for the other two integration ranges is not easy to understand, as it cannot be directly linked to a more or less strict selection of a specific molecular orientation.

For this reason, we prefer to keep the original presentation in the revised version of the manuscript.

Our critique so far mainly concerns statistical uncertainties. (We believe there should be 3-sigma confidence, at least, to make a claim). Quite obviously, there are also systematic factors – in the experiment as well as in the theoretical model. Our conclusion is that there are indications that the polarization of the CO2 molecular ion causes a delay on photoelectron emission (which is hardly surprising, but potentially interesting), but there is not yet evidence, not to speak of proof.

We would like to point out that the experimental error bars are mostly limited by the current technological limitations of these experiments. Our experimental system can perform attosecond photoelectron interferometry coincidence measurements for extremely long acquisition times (up to 76 hours), which are among the longest reported in the literature. These extended acquisition times are dictated by the coincidence technique used in the experiment and essentially determine the number of counts acquired in the experiment and therefore the error bars.

Regarding the experimental data, we believe that the clear shift of the experimental points and the good agreement with the theoretical data provide solid evidence that the effect of ionic coupling between the B and C states was observed in the experiment.

We also have a number of remarks that may be of more technical nature:

- The spectrum Fig 1c shows features not explained in the text, in particular the switching of the peaks from the green to the red line. What is the explanation and what are the implications for the experiment and its interpretation.

The green dotted and red dashed lines indicate the expected position of the photoelectron peaks generated by single XUV photon absorption corresponding to an ion in the final state $X^2\Pi_g$ and $B^2\Sigma_u^+$, respectively, as indicated in the caption. The reason why the total photoelectron spectrum shows peaks that are better aligned with the green lines in the low energy region (0-6 eV) and with the red

lines in the high energy region (6-16 eV) is due to the variations in the photoionisation cross sections for these final ionic states in this energy range, and to the XUV spectral distribution.

In fact, the adjacent green and red lines are due to the absorption of photons from different harmonic orders, due to the different ionisation potential of the two states (see Fig. 1b).

These observations are not crucial for the interpretation of the experiments, since our analysis focused on the blue shaded regions and not on the main peaks of the photoelectron spectra.

- The separation of the lines in Fig 1c does not match the photon energy, it is not even constant.

We thank the reviewer for this observation and apologise for overlooking this problem. In the revised version of the manuscript, we have provided the lines with the correct energy spacing.

The non-uniform spacing was due to an error in the calibration of the high-energy part of the photoelectron spectra. In addition to Fig. 1c, we have now also corrected the energy calibration in Fig. 2a,b,c,d, Fig.3a,b, Extended Data Fig. 1 and Extended Data Fig. 3.

- The text states that "the experimental data clearly show a linear variation". We clearly see a curvature. Is this reproduced by theory?

We thank the reviewer for this comment. We modified the text accordingly (page 8 line 150):

The experimental data clearly show a variation with the photoelectron kinetic energy due to the attosecond chirp and the photoionization molecular phase.

Reviewer #4 (Remarks to the Author):

We are grateful to the reviewer for her/his time and effort in the review process of our manuscript.

Additional Changes:

Furthermore, in the revised version of the manuscript, we have replaced the dipole operator in the ion with \mathbf{D} instead of \mathbf{d} , and have therefore also harmonised the notation in the supplementary material.

For the same reason, we have adapted the notation in Fig. 4.

Finally, we have corrected minor typos in the manuscript and added a few acknowledgments in the dedicated section.

We thank all the reviewers for their comments, which helped us to strengthen and clarify several points of our presentation. We hope that this revised version of the manuscript will be considered suitable for publication in Nature Communications.

On behalf of the co-authors,
Sincerely

Giuseppe Sansone

Point-by-point response for the manuscript:

Entanglement in photoionisation reveals the effect of ionic coupling in attosecond time delays
Nature Communications manuscript NCOMMS-24-81266-A

We would like to thank the reviewers for their remarks. According to their indications and suggestions, we have prepared a revised version of the manuscript. Hereafter, we include our point-by-point response to their comments. We have adopted the following color-code to simplify the presentation of our answers and to illustrate the changes introduced in the revised manuscript:

- **In black**, we report the original comment/remark of the reviewer:
- **In red**, we highlight our answers.
- **In blue**, we indicate the main changes introduced in the revised version of the manuscript.

In the resubmission, we provide two versions of the main manuscript and supplementary information. In the first, we have highlighted all major changes in the revised version of our paper, while the second is a clean version of the revised work.

The pages and lines used to indicate the location of the changes discussed in this point-by-point response refer to the version of the manuscript in which the major changes are highlighted.

Reviewer #1 (Remarks to the Author):

I was the Reviewer #1 for the manuscript "Entanglement in photoionisation reveals the effect of ionic coupling in attosecond time delays." This is my second review. My first review contained six comments, and I appreciate the efforts by the authors to address them. Here are my responses to the revision by the authors.

(1) The revised introduction gives better explanations of the RABBIT technique. I appreciate the efforts by the authors. There is no more question from my end.

(2) The added annotations in Figure 3 are helpful, but the figure still looks congested. In my personal opinion, plotting six curves in a single figure (Fig. 3(b)) is not a good idea. There is no more question from my end, though.

(3) My comment #3 was about the distinction of correlation and entanglement. The response by the authors is difficult for me to follow.

(3-1) It remains unclear to me, in the model, how we can compare the classical ion-electron pairs and the entangled ion-electron pairs. If with/without BC coupling is taken equivalent to entangled/classical pairs, that I think should be clearly stated so.

In our model, photoelectrons/photoions resulting from XUV photoionisation are entangled even in the absence of BC coupling. Therefore, the entangled nature of the photoelectron/photoion is unaffected by the presence or absence of BC coupling.

However, BC coupling enables an additional attosecond time delay to be observed, depending on the entanglement between the photoelectron and photoion. This is summarised in Eq. (7) in the first revision of the manuscript, where we demonstrate that the additional attosecond time delay depends on BC coupling (D_{BC}) and the function $X(\gamma)$, which is dependent on the purity of Ψ_{2q+1} (see Eq. (4)). Consequently, the delay observed in the experiment is determined by the BC coupling and the degree of entanglement.

(3-2) I understand that the additional photoionization delay is the key information about the entanglement, and that corresponds to the deviation between the black curve (th. CC) and the blue curve (th. no B-C) in Fig. 3(a). I need to point out that the deviation and the experimental error bars are about the same amplitude. I would like to know how much confidence the authors have about the importance of the B-C coupling when compared to the experimental data.

To answer this question raised by the reviewer, we include here the deviations (in unit of the standard deviation) and the corresponding confidence values for the the two data points at 28.2 eV and 31 eV

between the experimental data and the model without B-C coupling for the different integration angles:

Energy (eV)	τ_{exp} (as)	σ_{exp} (as)	τ_{theory} (as)	Z-score $ \tau_{exp} - \tau_{theory} /\sigma_{exp}$	Confidence level
0° – 20°					
28.98	-131.1	31.8	-43.72	2.75	0.6%
31.42	-169.34	41.27	-67.51	2.47	1.35%
0° – 50°					
28.98	-116.9	30.12	-39.59	2.57	1.02%
31.42	-150.74	34.73	-57.85	2.68	0.74%
0° – 90°					
28.98	-86.9	29.69	-32.56	1.83	6.72%
31.42	-100.04	34.23	-46.36	1.57	11.64%

In particular, for the data with smaller integration angles (i.e. for the conditions under which the effect of the B-C coupling is the most relevant) the deviation is more than $2.47\sigma_{exp}$ corresponding to a confidence of only 1.35%. For the total integration angle the deviation is on the order of $1.57\sigma_{exp}$ and $1.8\sigma_{exp}$ corresponding to confidence intervals of 11.64% and 6.72%.

Our conclusion is that the deviation in unit of standard deviation (or equivalently the confidence level) indicates that our experimental data are not compatible with the predictions of the full model without B-C coupling. This conclusion is particularly supported by the analysis of the angle resolved spectra for which deviations close or greater than $2.5\sigma_{exp}$ are observed. To highlight the deviation from the model without B-C coupling, we have added the following sentence to the manuscript (page 10 lines 197-199).

The comparison between the experimental points and the two theoretical model with and without B-C coupling for the integration angles 0° – 20° and 0° – 50° is presented in the Extended Data Fig. 6.

In the supplementary Material we have added Extended Data Fig. 6 presenting the comparison and this additional description (page 4 line 28- page 5 line 36):

The photoionisation time delays $\tau = \tau_B - \tau_A$ were obtained as the difference of the delays derived from RABBIT traces measured in coincidence with CO_2^+ and Ar^+ . The comparison between the experimental time delays τ and those obtained from the full model and the model without B-C coupling for RABBIT traces integrated over the photoelectron emission angle (0° – 20°) and (0° – 50°) are presented in Fig. Extended Data Fig. 6a and b, respectively. For the experimental points at 28.98 eV and 31.42 eV, the deviation from the model without B-C model is 2.75σ and 2.47σ , 2.57σ and 2.68σ , and 1.83σ and 1.57σ for the integration angles 0° – 20°, 0° – 50°, and 0° – 90°, respectively (where σ indicates the corresponding standard deviation for each experimental point).

(4) The new estimate of the purity is useful and I appreciate the efforts by the authors to include this number.

The lines 274-275 “The purity ... is about 0.5-0.6” make a strange phrase. There should be just one number and a standard deviation, not a range.

We have replaced this sentence with the following one (page 14, lines 276-277):

The average purity γ in the photon energy range between 22 eV and 40 eV is 0.53 ± 0.02

(5) The newly added text is helpful. My original question arose because “the observation that the contribution does not depend on the distance between the photoelectron and photoion” sounded experimental, and I wanted to see such data. But now it seems that this is all within the theoretical model and there is no experimental evidence. There is no more question from my end.

(6) The typos have been nicely fixed. There is no more question from my end.

Overall, I think that the quality of experimental and theoretical data in the manuscript is about the average of articles in Nature Communications, and this can be a good stimulating paper for the

community. My reservation is that the level of confidence in overall arguments needs to be carefully adjusted, especially around the role of entanglement. For example, changing the title (also suggested by the reviewer #2) seems appropriate. In my personal view, this paper can be one of the early works that suggest or try to address the potential implication of attosecond entanglement. I envision that more future works will follow with improved experimental design and statistics to establish the experimental manifestation of entanglement in attosecond experiments.

We thank the reviewer for their thoughtful feedback and for recognizing the potential of our work to stimulate further research in the field. While we appreciate the suggestion to revise the title, we respectfully believe that the current title reflects the content and conclusions of our study. We point out that ion-photoelectron entanglement alone does not induce an additional delay. However, entanglement is necessary in order to induce an additional delay originating from IR-induced BC coupling.

Additionally, Reviewer #2, who initially suggested this change, has been convinced by our revisions and no longer recommends altering the title.

Reviewer #2 (Remarks to the Author):

The authors have fully addressed all my previous comments in a detailed and convincing way. I also believe that, in the revised manuscript, the discussion on the connection between entanglement and photoionization time delay is certainly more complete and rigorous than it was before. Therefore, I am happy to recommend the work "Entanglement in photoionisation reveals the effect of ionic coupling in attosecond time delays" for publication in Nature Communications in the current form.

We thank the reviewer for the kind feedback and are pleased that the revisions addressed all comments satisfactorily. We appreciate the recommendation for publication.

Reviewer #3 (Remarks to the Author):

The central critique of our first report have been doubts about the significance of the results in the sense of statistical and systematic errors. We had asked what the confidence level is, particularly for the data points at 28.2eV and 31eV in Fig 3a. This question, which in our opinion is a crucial one in physics, is not addressed in the authors' answer, not to say that it is being evaded.

The authors claim that the strong deviation of the data point at 24eV is due to "limitations of the theoretical model in accurately describing the electronic states within this region", where "this" refers to the vicinity of the ionization threshold. However, the C state has a binding energy of 19.4eV, almost 5eV less. In addition, the points closer to the ionization threshold fit better. Moreover, the paper claims that the model "simulates the correlations with high accuracy" (l. 174). What is to be understood by „high accuracy“?

The "high accuracy" mentioned in the manuscript refers to the excellent agreement between theoretical photoionisation cross sections obtained from the discussed close-coupling R-matrix model and the available experimental data on this process. The correlation has been shown in earlier works (all cited in the manuscript) as crucial to recover the shape of major resonances in the cross sections, although the positions are not always spot on with the experiment.

We have added the following additional text in the manuscript (page 9 lines 172-176):

in the initial bound state as well as in the final continuum states. The adequacy of the employed R-matrix model has been demonstrated in theoretical treatment of one-photon ionization^{54, 55}, where it yielded photoionization cross sections in good agreement with measurements, including the shapes of major resonances.

The mentioned "limitations" refer particularly to the high amount of minor autoionising resonances in the vicinity of thresholds. While the measurements are performed at energies above the C channel threshold, the calculation also contains a few hundred other thresholds (describing approximately the

Rydberg states of the ion converging to various thresholds). These have more of a numerical role in the calculation and their thresholds are hard to converge to actual physical thresholds. The position of an autoionising resonance associated with such a state is necessarily subject to a corresponding uncertainty in energy. We attribute the offset of the 24 eV experimental data point away from the theoretical prediction to effects occurring in the C state continuum which includes the formation of an autoionizing resonance. We included a discussion of this aspect in the Supplementary Material.

In our opinion, two things are mandatory before publication can be considered

- It MUST be shown that the deviation of the data points at 28.2eV and 31eV from the non-complete models is significant with at least 3 sigma (or another criterion that is similarly convincing)

Following the reviewer recommendation, we include here the deviations (in unit of the standard deviation) and the corresponding confidence values for the the two data points at 28.2 eV and 31 eV between the experimental data and the model without B-C coupling.

Energy (eV)	τ_{exp} (as)	σ_{exp} (as)	τ_{theory} (as)	Z-score $ \tau_{exp} - \tau_{theory} /\sigma_{exp}$	Confidence level
0° – 20°					
28.98	-131.1	31.8	-43.72	2.75	0.6%
31.42	-169.34	41.27	-67.51	2.47	1.35%
0° – 50°					
28.98	-116.9	30.12	-39.59	2.57	1.02%
31.42	-150.74	34.73	-57.85	2.68	0.74%
0° – 90°					
28.98	-86.9	29.69	-32.56	1.83	6.72%
31.42	-100.04	34.23	-46.36	1.57	11.64%

In particular, for the data with smaller integration angles (i.e. for the conditions under which the effect of the B-C coupling is the most relevant) the deviation is more than $2.47\sigma_{exp}$ corresponding to a confidence of 1.35%. For the total integration angle the deviation is on the order of $1.57\sigma_{exp}$ and $1.8\sigma_{exp}$ corresponding to confidence intervals of 11.64% and 6.72%.

Our conclusion is that the deviation in unit of standard deviation (or equivalently the confidence level) indicates that our experimental data are not compatible with the predictions of the full model without B-C coupling. This conclusion is particularly supported by the analysis of the angle resolved spectra for which deviations close or greater than $2.5\sigma_{exp}$ are observed. To highlight the deviation from the model without B-C coupling, we have added the following sentence to the manuscript (page 10 lines 197-199).

The comparison between the experimental points and the two theoretical model with and without B-C coupling for the integration angles 0° – 20° and 0° – 50° is presented in the Extended Data Fig. 6.

In the supplementary Material we have added Extended Data Fig. 6 presenting the comparison and this additional description (page 4 line 28- page 5 line 36):

The photoionisation time delays $\tau = \tau_B - \tau_{Ar}$ were obtained as the difference of the delays derived from RABBIT traces measured in coincidence with CO_2^+ and Ar^+ . The comparison between the experimental time delays τ and those obtained from the full model and the model without B-C coupling for RABBIT traces integrated over the photoelectron emission angle (0° – 20°) and (0° – 50°) are presented in Fig. Extended Data Fig. 6a and b, respectively. For the experimental points at 28.98 eV and 31.42 eV, the deviation from the model without B-C model is 2.75σ and 2.47σ , 2.57σ and 2.68σ , and 1.83σ and 1.57σ for the integration angles 0° – 20°, 0° – 50°, and 0° – 90°, respectively (where σ indicates the corresponding standard deviation for each experimental point).

Moreover, we would like to point out that, in attosecond science, it is widely accepted to evaluate agreement (or disagreement) at the level of one standard deviation (1σ) or two standard deviations

(2σ). This approach has been adopted and reported in numerous key publications, including the following:

1. Kotur, M. *et al.* Spectral phase measurement of a Fano resonance using tunable attosecond pulses. *Nat Commun* **7**, 10566; 10.1038/ncomms10566 (2016).
2. Isinger, M. *et al.* Photoionization in the time and frequency domain. *Science* **358**, 893–896; 10.1126/science.aao7043 (2017).
3. Busto, D. *et al.* Fano's Propensity Rule in Angle-Resolved Attosecond Pump-Probe Photoionization. *Phys. Rev. Lett.* **123**, 133201; 10.1103/PhysRevLett.123.133201 (2019).
4. Zhong, S. *et al.* Attosecond electron-spin dynamics in Xe 4d photoionization. *Nat Commun* **11**, 5042; 10.1038/s41467-020-18847-1 (2020).
5. Nandi, S. *et al.* Attosecond timing of electron emission from a molecular shape resonance. *Science advances* **6**, eaba7762; 10.1126/sciadv.aba7762 (2020).
6. Lucchini, M. *et al.* Unravelling the intertwined atomic and bulk nature of localised excitons by attosecond spectroscopy. *Nat Commun* **12**, 1021 (2021).
7. Potamianos, D. *et al.* Attosecond chronoscopy of the photoemission near a bandgap of a single-element layered dielectric *Science advances* **10**, eado0073 (2024).

For this reason, we believe that the discrepancy between our experimental results and theoretical predictions, which does not include B-C coupling at a level of roughly 2.5 sigma (at least for the two smaller integration angles), is strong evidence that our experimental data are not compatible with the model without B-C coupling. This provides solid experimental evidence of the role played by B-C coupling.

- It MUST be explained convincingly why the model(s) deviate strongly from the experiment at 24eV and why the models are nevertheless reliable at the other energies.

We attribute the offset of this point to a small shift of the positions of autoionizing resonances in CO₂ with respect to their accurate experimental values. Indeed, a comparison of the experimental and simulated cross sections and asymmetry parameters for the C state reveals an upshift of approximately 2–3 eV in the energy position of the simulated autoionising resonances of this state, compared to the values reported in the literature. The IR-induced transition between the C and B states establishes a link between the characteristics of the C-state single XUV-photon continuum and the two-photon (XUV and IR) delays measured in the B-state. For this reason, a comparable energy shift is expected for the photoionisation time delays τ_B . To support this argument qualitatively, we present the downshifted RABBIT delay $\tau = \tau_B - \tau_{Ar}$ in this answer, for which different energy shifts ($\Delta E = 0, -2.0, -2.5, -3.0$ eV) have been applied to the delay τ_B . The position of the autoionising resonances in argon is correct, so they do not need to be shifted.

Figure 1 Comparison between the experimental point at 24.19 eV and the RABBIT delay τ , obtained considering different energy shifts of the delay τ_B : $\Delta E = 0$ eV (grey), $\Delta E = -2.0$ eV (red), $\Delta E = -2.5$ eV (blue), and $\Delta E = -3.0$ eV (green).

As can be seen, there is now strong qualitative agreement between the experimental point at 24.19 eV and the model prediction.

We have provided a detailed explanation of this argument in the revised supplementary material. In particular, we have added a section to the supplementary material entitled '**Role of resonances and structured continua around 24 eV**' (pages 6-8 of the supplementary material). Furthermore, we have included a supplementary figure (Extended Data Fig. 7) to illustrate our arguments.

In the main text of the manuscript, we have included the following sentence (page 10, lines 194-197): We attribute the offset of this point to a small shift of the positions of autoionizing resonances in CO_2 with respect to their accurate experimental values. See SM 1.3 and Extended Data Fig. 7 for a detailed discussion.

Finally, we (and reviewer 1) ask ourselves whether framing the work in terms of entanglement helps in elucidating anything. After all, it has always been obvious (or, at least, should have been obvious) that the fragments of photoionization and -dissociation are entangled. Is it just new language (new in the attosecond community) or does it provide insight that may help to advance the field?

As we indicated in our response to Reviewer 1, we believe that Eq. (7) provides insight into the significance of entanglement, as it demonstrates the qualitative and quantitative impact of the degree of entanglement (as measured by the purity parameter, γ) on the additional attosecond time delay through the function $X(\gamma)$. Furthermore, the supplementary material reports the full dependence of $X(\gamma)$ on the purity γ .

Finally, we do not believe that our observations in CO_2 are a special case, but rather that similar resonances can occur in different molecules. Therefore, our findings and their interpretation in terms of entanglement are likely to be relevant to the community working on RABBIT measurements in molecular systems.

We are also a bit puzzled about the role of the IR field: It affects the properties of the photoelectron and, at the same time, is used to measure the delay. It seems that both roles are, well, entangled. A natural specific question in this respect would be if the measured delay depends on the IR field strength?

The fact that the IR measures and simultaneously 'affects' the system is well known in attosecond physics. In particular, interaction with the IR introduces an additional delay, typically referred to as continuum-continuum delay in the case of photoionisation from atoms. There has been a great deal of research into characterising and isolating the continuum-continuum delay in measured attosecond time delays (see, for example: Dahlstrom, J. M., L'Huillier, A. & Maquet, A. *Introduction to attosecond delays in photoionization*. Journal of Physics B: Atomic, Molecular and Optical Physics 45, 183001 (2012); Dahlstrom, J. et al. *Theory of attosecond delays in laser-assisted photoionization*. Chemical Physics 414, 53–64 (2013); Pazourek, R., Nagele, S. & Burgdorfer, J. *Attosecond chronoscopy of photoemission*. Reviews of Modern Physics 87, 765–802 (2015)).

In general, the attosecond time delay in photoionisation depends on the strength of the IR field. However, understanding and evaluating its evolution with the IR field is not simple, as it requires the inclusion of perturbation theory considering the exchange of more than one IR photon with the external field.

Nevertheless, the experiment is performed at such low IR intensities (typically 5×10^{11} W/cm²) that only single-photon NIR transitions are significant. This is consistent with the adopted model, which is based on a perturbative approach that only considers the exchange of a single NIR photon with the external field.

Reviewer #4 (Remarks to the Author):

We are grateful to the reviewer for her/his time and effort in the review process of our manuscript.

On behalf of the co-authors,
Sincerely

Giuseppe Sansone

Point-by-point response for the manuscript:

Entanglement in photoionisation reveals the effect of ionic coupling in attosecond time delays
Nature Communications manuscript NCOMMS-24-81266-B

We would like to thank the reviewers for their remarks. According to their indications and suggestions, we have prepared a revised version of the manuscript. Hereafter, we include our point-by-point response to their comments. We have adopted the following color-code to simplify the presentation of our answers and to illustrate the changes introduced in the revised manuscript:

- **In black**, we report the original comment/remark of the reviewer:
- **In red**, we highlight our answers.
- **In blue**, we indicate the main changes introduced in the revised version of the manuscript.

In the resubmission, we provide two versions of the main manuscript and supplementary information. In the first, we have highlighted all major changes in the revised version of our paper, while the second is a clean version of the revised work.

The pages and lines used to indicate the location of the changes discussed in this point-by-point response refer to the version of the manuscript in which the major changes are highlighted.

Reviewer #1 (Remarks to the Author):

The response letter contained some useful information, but it was not added to the main manuscript. Adding the following two sentences will help make the work accessible to the broad readership of Nat Commun:

“In our model, photoelectrons/photoions resulting from XUV photoionisation are entangled even in the absence of BC coupling. Therefore, the entangled nature of the photoelectron/photoion is unaffected by the presence or absence of BC coupling. However, BC coupling enables an additional attosecond time delay to be observed, depending on the entanglement between the photoelectron and photoion.”

“We point out that ion-photoelectron entanglement alone does not induce an additional delay. However, entanglement is necessary in order to induce an additional delay originating from IR-induced BC coupling.”

We thank the reviewer for these suggestions. We have added the following sentences in the manuscript:

In our model, photoelectrons/photoions resulting from XUV photoionisation are entangled even in the absence of the coupling between the ionic states. Therefore, the entangled nature of the photoelectron/photoion is unaffected by the presence or absence of this additional coupling. However, the coupling enables an additional attosecond time delay to be observed, depending on the entanglement between the photoelectron and photoion. We point out that ion-photoelectron entanglement alone does not induce an additional delay. However, entanglement is necessary in order to observe an additional delay originating from IR-induced ionic coupling.

Reviewer #3 (Remarks to the Author):

The authors have carefully considered our concerns. Their analysis indicates a sufficient confidence of the validity of their conclusions.

As a remark: It is true that it is general practice, not only in attoscience, to indicate 1-sigma confidence intervals on data points – like in, e.g., [2] Isinger et al. To illustrate our point: If there had been only

one gray diamond data point in their Fig 1A, the case would have been weak. However, there are five such points, all deviating more than 1 sigma in the same direction.

In the present case, there 2 times 3 data points, however they use partially the same events. On the other hand, two independent points with a 1% probability that they are compatible with the negative hypothesis are satisfactory.

We thank the reviewer for the positive feedback.

Reviewer #4 (Remarks to the Author):

We are grateful to the reviewer for her/his time and effort in the review process of our manuscript.

On behalf of the co-authors,
Sincerely

Giuseppe Sansone